# Long lifetimes white afterglow in slightly crosslinked polymer systems

Qingao Chen[1,3], Lunjun Qu [1,3], Hui Hou[1], Jiayue Huang[1], Chen Li[1], Ying Zhu[1], Yongkang Wang[1], Xiaohong Chen[1], Qian Zhou[1], Yan Yang[1] & Chaolong Yang [1,2] ✉

Intrinsic polymer room-temperature phosphorescence (IPRTP) materials have attracted considerable attention for application in flexible electronics, information encryption, lighting displays, and other fields due to their excellent processabilities and luminescence properties. However, achieving multicolor long-lived luminescence, particularly white afterglow, in undoped polymers is challenging. Herein, we propose a strategy of covalently coupling different conjugated chromophores with poly(acrylic acid (AA)-AA-N-succinimide ester) (PAA-NHS) by a simple and rapid one-pot reaction to obtain pure polymers with long-lived RTPs of various colors. Among these polymers, the highest phosphorescence quantum yield of PAPHE reaches 14.7%. Furthermore, the afterglow colors of polymers can be modulated from blue to red by introducing three chromophores into them. Importantly, the acquired polymer TPAP-514 exhibits a white afterglow at room temperature with the chromaticity coordinates (0.33, 0.33) when the ratio of chromophores reaches a suitable value owing to the three-primary-color mechanism. Systematic studies prove that the emission comes from the superposition of different triplet excited states of the three components. Moreover, the potential applications of the obtained polymers in light-emitting diodes and dynamic anti-counterfeiting are explored. The proposed strategy provides a new idea for constructing intrinsic polymers with diverse white-light emission RTPs.

Pure organic room-temperature phosphorescence (RTP) materials have attracted extensive attention for application in multiple anti-counterfeiting encryption[1–3], biological imaging[4], and organic optoelectronic devices[5] due to their easy processings, low costs, and convenient functionalizations[6]. However, generating stable triplet excitons in these materials is challenging because of the difficult intersystem crossing (ISC) from the lowest excited singlet state ($S_1$) to the triplet state ($T_n$) with low spin-orbit coupling (SOC) constant and probability of easy quenching of triplet excitons by water or oxygen in these materials[7]. As radiative transitions of excitons from the triplet excited state ($T_1$) to the singlet ground state ($S_0$) are spin-forbidden, achieving organic RTP systems with high luminous efficiencies is challenging[8]. In the past few years, a series of strategies, such as introduction of heavy atoms[9], heteroatoms[10], carbonyls[11], and halogens[12], have been used to enhance the SOCs of organic systems for promoting the probability of ISC. Additionally, rigid environments have been constructed by H-aggregation[13], crystallization engineering[14,15], and host–guest doping[16–18] to suppress non-radiative transitions of triplet excitons. Nevertheless, developing a strategy for the synthesis of efficient and stable organic RTP materials still remains a challenge.

[1]School of Materials Science and Engineering, Chongqing University of Technology, Chongqing 400054, China. [2]Guangdong Provincial Key Laboratory of Luminescence from Molecular Aggregates, South China University of Technology, Guangzhou 510640, China. [3]These authors contributed equally: Qingao Chen, Lunjun Qu. ✉e-mail: yclzjun@163.com

Large molecular-weight polymers that can offer rigid environments and abundant interaction forces to inhibit the non-radiative transitions of triplet excitons have attracted considerable interest from researchers owing to their low costs, low toxicities, excellent processabilities, and modifiabilities[19–25]. In previous studies, doping small organic molecules into rigid polymer matrices by physical mixing is used as a common strategy for obtaining polymers with long-lived RTPs[19,26–29]. However, due to phase separation in the doping system, the polymer matrix easily becomes exposed to the aqueous environment, leading to phosphorescence quenching. To solve this problem, more stable undoped polymeric RTP materials based on covalent bonds have received extensive attention[30,31]. However, complicated preparations and narrow application ranges have significantly hindered the construction of these polymeric materials. Therefore, the development of more convenient strategies for the synthesis of IPRTP materials with high afterglow properties is important.

In recent years, compared with conventional RTP materials with only single afterglow colors, polymers with dynamically tunable multicolor phosphorescences have become more attractive[32,33]. Constructing multiple emission centers in a polymer is an important method to acquire multicolor phosphorescence. Zhao et al.[34] conjugated multiple emission centers to the main chain of a polymer via free-radical crosslinking copolymerization to obtain an organic material that demonstrates ultralong RTP in response to the excitation wavelength. Yuan et al.[35] grafted different luminescent units onto sodium alginate chain to produce amorphous polymers with different RTPs. Furthermore, multicomponent hybrid white RTP systems can be fabricated by completely utilizing triplet energy, which exhibit more attractive application prospects in the fields of lighting and display[36]. Based on the construction of an effective Förster resonance energy transfer (FRET) system[37], researchers have achieved white-light emission in pure organic systems via $S_1$-$S_1$ mixing[38], $S_1$-$T_1$ mixing[39], and $T_1$-$T_1$ mixing[40]. Nevertheless, to date, the

construction of IPRTP materials with white afterglow has rarely been reported.

Herein, we provide a convenient and effective strategy for the synthesis of intrinsic polymers with color-tunable, long-lived RTPs. At first, the copolymer polyacrylic acid (PAA)-acrylic acid (AA)-N-succinimide ester (PAA-NHS) was fabricated via free-radical polymerization[41,42]. In this system, the lone pair electrons of the N atom on the amino group can interact with the π structure, inducing an n→π* transition and promoting ISC from $S_n$ to $T_n$ that produces numerous triplet excitons[43]. Furthermore, both the rich H bonds formed between the abundant carboxyl and carbonyl groups on the polymer side chains[44] and long-chain winding structure of the polymer can harden the molecular conformation and improve the rigidity of the system, thereby inhibiting non-radiative relaxation. Subsequently, four types of phosphorescent chromophores with different degrees of conjugation were simultaneously grafted onto PAA-NHS by a convenient one-pot method in a short time because of the high reactivities of electron-rich amino groups and NHS esters. Afterglow of the new polymers were dynamically adjusted from blue to red by modulating the proportions of different chromophores. Finally, when the proportions of phosphorescent chromophores reached suitable values, the obtained slightly crosslinked polymers exhibited attractive white afterglow, which demonstrated excellent applications in light-emitting diodes (LEDs) and dynamic anti-counterfeiting. This study offers a new approach for constructing RTP platforms in intrinsic polymer systems.

## Results

To acquire intrinsic polymer systems with long-lived white afterglow, initially, a simple copolymer (PAA-NHS) was prepared by free-radical copolymerization of AA and AA-N-succinimide ester (AA-NHS) at an optimum molar feed ratio of 70:1 using azobisisobutyronitrile (AIBN) as the initiator and dimethyl sulfoxide (DMSO) as the solvent (Fig. 1). Then, four conjugated chromophores, namely, 2-(3-aminophenyl)-5-

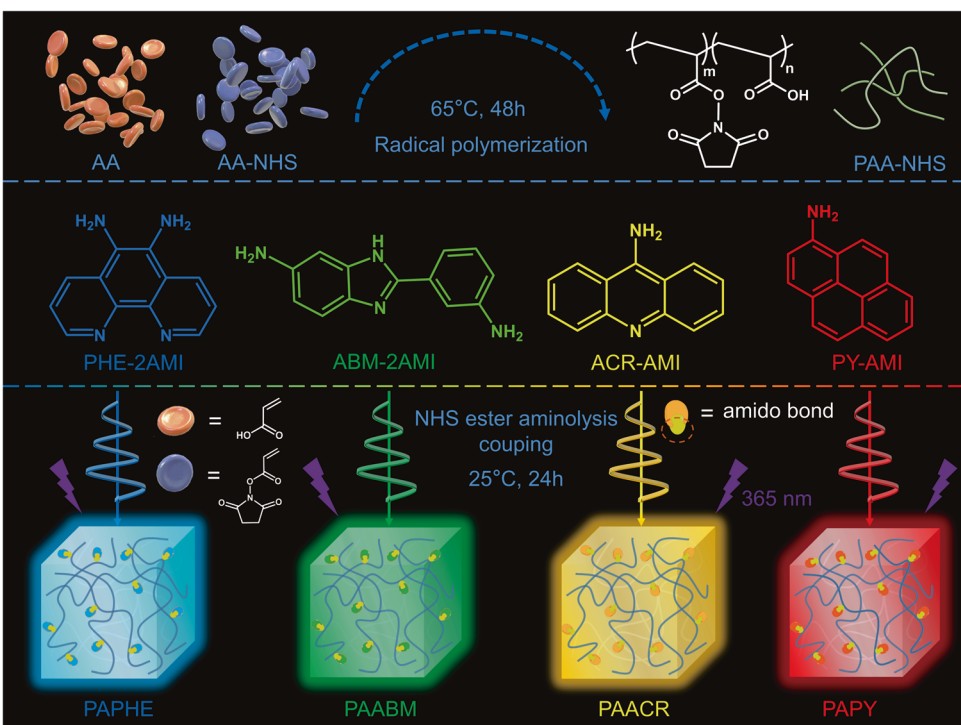

**Fig. 1 | The strategy diagram of one-pot preparation of long-lived amorphous polymeric RTP systems.** The top part is the process of synthesizing the polymer precursor PAA-NHS by free radical copolymerization of acrylic acid and acrylic acid-N-succinimide ester. 'm' and 'n' represent the copolymerization ratio of monomer

AA and AA-NHS. The middle part is the chemical formula of four small molecules as luminescent centers. The following section shows the process of generating intrinsic polymer RTP materials by the reaction of small molecules with PAA-NHS through aminolysis coupling. Lightning represents 365 nm UV excitation source.

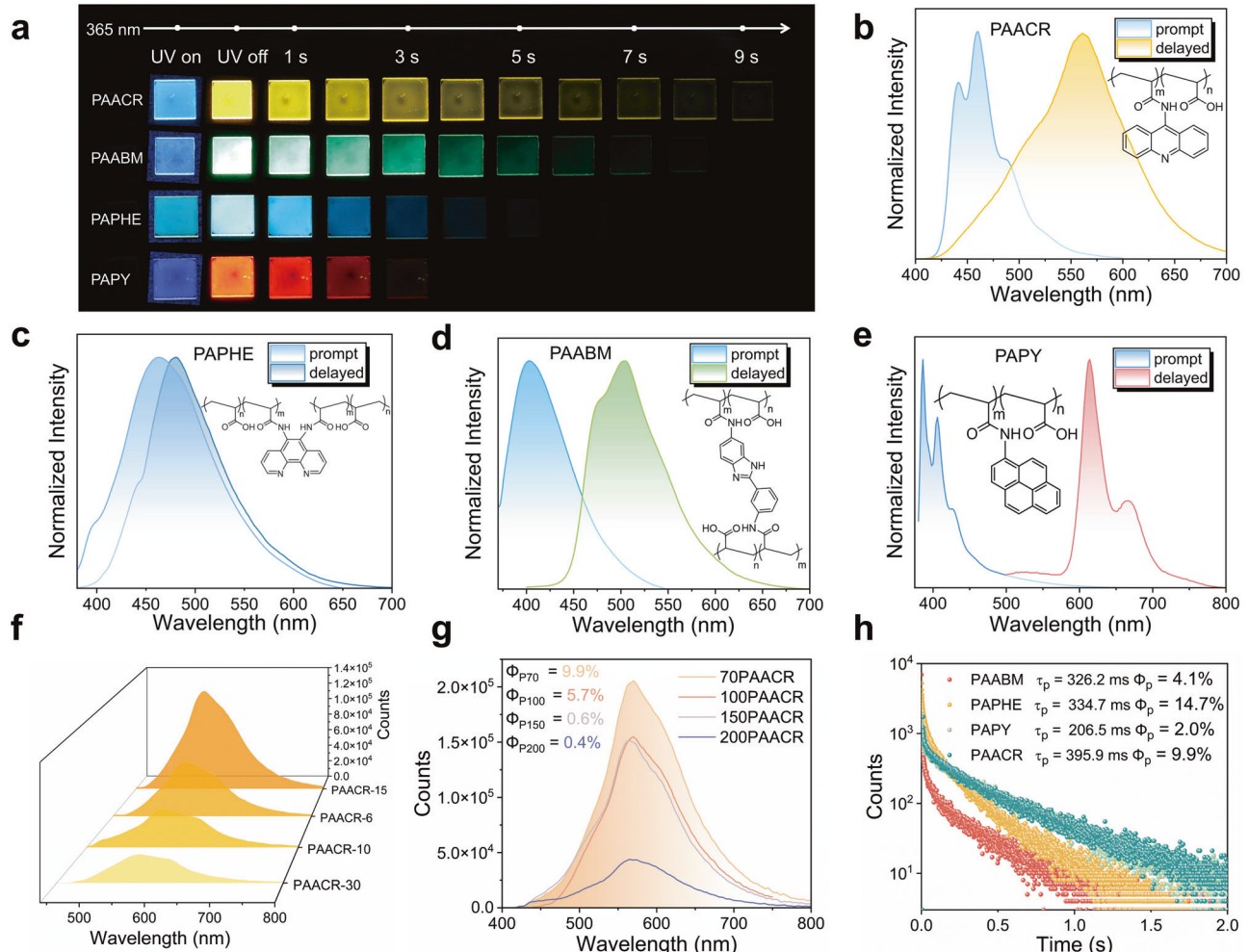

**Fig. 2 | Photophysical properties of single-component polymer films with room-temperature phosphorescences. a** Images of long-lived RTPs of four single-component polymer films excited by a 365 nm UV flashlight. Photoluminescence spectra of PAACR (**b**), PAPHE (**c**), PAABM (**d**), and PAPY (**e**) acquired under 365 nm excitation and ambient conditions. **f** Phosphorescence spectra of PAACR films with different chromophore concentrations under 365 nm excitation at room temperature. **g** Phosphorescence spectra and quantum yields of PAACR films with different acrylic acid contents under 365 nm excitation at room temperature. **h** Phosphorescence lifetimes and quantum yields of the four single-component polymer films under 365 nm excitation.

aminobenzimidazole (ABM-2AMI), 5,6-diamino-1,10-phenanthroline (PHE-2AMI), 9-aminoacridine (ACR-AMI), and 1-aminopyrene (PY-AMI), were introduced into the reaction system to react with PAA-NHS for a short time at room temperature; the active NHS ester covalently coupled with the amino groups to form stable amide bonds. Finally, PAPHE, PAABM, PAACR, and PAPY were achieved, purified by reprecipitation, and then prepared into thin films by coating (Supplementary Fig. 1).

Chemical structures of these products were characterized by proton nuclear magnetic resonance ($^1$H, $^{13}$C NMR) spectra (Supplementary Figs. 2–10). The absence of amino proton peaks and presence of phenyl proton peaks in the spectra of the products indicated that the phosphor molecules were successfully grafted onto PAA-NHS. As the amounts of the grafted phosphorescent chromophores were very small, no significant absorption vibration peak of the benzene ring was observed in the Fourier transform infrared (FT-IR) spectra of the products (Supplementary Fig. 17). Gel permeation chromatography revealed that the maximum average molecular weight ($M_n$) of PAPHE was 44248 g/mol (Supplementary Fig. 18), which was conducive to the formation of flexible films. Additionally, X-ray diffraction (XRD) results of these polymer films exhibited small and wide diffraction peaks, indicating their amorphous structures (Supplementary Fig. 19). More importantly, the rigid PAA chains endowed these polymer films with

appropriate thermal stabilities, as confirmed by thermogravimetric (TG) and differential scanning calorimetry (DSC) analyses (Supplementary Figs. 20, 21), indicating against certain thermal aging in the environment.

As expected, long-lived IPRTP systems were effectively fabricated by grafting the four organic phosphors onto PAA-NHS. Ultraviolet-Visible absorption spectra of the prepared polymer films indicated significant π → π* and n→π* transitions (Supplementary Fig. 22). After the removal of 365 nm UV irradiation, these polymer films exhibited intense afterglow ranging from blue to red (Fig. 2a). Taking ACR-AMI as an example, the corresponding polymer film is termed PAACR. Contents of conjugated chromophores substantially affect the luminescence properties of the polymer[45]. Subsequently, a series of polymer films, PAACR-6, PAACR-10, PAACR-15, and PAACR-30, were prepared using the ACR-AMI molar feed ratios of 0.06, 0.10, 0.15, and 0.30, respectively. Phosphorescence intensities of the polymer films at 566 nm increased at first and then decreased with an increase in the chromophore feed ratio (Fig. 2f), and the phosphorescence lifetimes also demonstrated the same variation trend (Supplementary Fig. 23). This is because the increase in the chromophore feed ratio can cause cluster aggregation that would aggravate intramolecular thermal motion of the chromophore, thereby enhancing non-radiative relaxation and leading to the decay of triplet excitons in a non-radiative

manner[46]. Influence of the content of AA on the phosphorescence properties of the polymer films 70PAACR, 100PAACR, 150PAACR, and 200PAACR with AA/AA-NHS ratios of 70/1, 100/1, 150/1, and 200/1, respectively, is further discussed. Phosphorescence intensities of the films gradually decreased with an increase in the AA content when other conditions were constant (Fig. 2g). Although the phosphorescence lifetimes of 100PAACR and 150PAACR were slightly higher than that of PAACR, the phosphorescence quantum yields were significantly lower (9.9%, 5.7% and 0.6% for PAACR and 100PAACR and 150PAACR, respectively) (Supplementary Fig. 24). Increase in the AA content can improve the rigidity of the polymer system and inhibit triplet exciton quenching, which can also increase the hygroscopicity and grafting difficulty of the polymer. Thus, the feed ratios of phosphorescent units in the polymer should be strictly controlled.

Subsequent experiments were conducted using the optimal molar feed ratio. Under environmental conditions, PAACR was excited by a 365 nm UV lamp, and it emitted bright blue fluorescence. After removing the UV lamp, PAACR exhibited yellow long-lived RTP for more than 9 s (Supplementary Movie 1). Steady-state and transient spectra measured under the environmental conditions appropriately corresponded with the observed results. Under 365 nm excitation, the phosphorescence lifetime recorded at 566 nm (yellow) was 395.9 ms for PAACR, with a fluorescence lifetime of 11 ns at 461 nm (blue) (Fig. 2b). Unexpectedly, the optimal excitation band of PAACR in the three-dimensional excitation–emission phosphorus spectrum ranged from 290 to 443 nm (Supplementary Fig. 25), demonstrating that PAACR can be excited in the visible light range (Supplementary Movie 2). Moreover, blue (475 nm), green (502 nm), and red (619 nm) afterglow with long lifetimes were successfully achieved for PAPHE, PAABM, and PAPY, respectively (Fig. 2c–e, Supplementary Movies 3–5). Time-resolved emission spectra (TRES) were further used to determine long phosphorescence lifetimes of the polymer films (Supplementary Fig. 26), which were 206.5 ms (PAPY), 362.2 ms (PAABM), and 334.7 ms (PAPHE). Notably, the phosphorescence lifetimes of PAPHE, PACCR, and PAABM are longer than that of PAPY. This may be because the N atoms in the aromatic rings of PAPHE, PACCR, and PAABM are more prone to form stronger H bonds with the polymer matrix when compared with the case of PAPY with a large conjugated pyrene ring structure. Furthermore, the slightly crosslinked polymer films PAABM and PAPHE stimulated the construction of a rigid environment, which resulted in longer phosphorescence lifetimes.

To quantitatively describe the photoluminescences (PL) of these polymer films, their phosphorescence quantum yields at room temperature were determined, which were 2.0% (PAPY), 4.1% (PAABM), 9.9% (PACCR), and 14.7% (PAPHE) (Fig. 2h). The corresponding color coordinates were obtained based on the International Commission on Illumination (CIE), which were in suitable agreement with the visual observation results (Supplementary Figs. 27, 28). Note that PAABM exhibited considerable excitation-dependent characteristics under 254 and 365 nm UV excitations (Supplementary Movie 6). The afterglow changed from green to blue cyan with a decrease in the excitation wavelength (Supplementary Fig. 29). Normalized phosphorescence spectra indicated that the maximum emission peak red-shifted with an increase in the excitation wavelength (Supplementary Fig. 30). Correspondingly, considerable linear changes were noticed in the CIE coordinates. This may be attributed to the different aggregation morphologies of the ABM-2AMI molecules in the polymer film. All photophysical data of the polymer system were systematically measured and are presented in Supplementary Table 2 and Supplementary Fig. 31. According to the literature[47], the emitting centers of polymers originate from phosphorescent small molecules. These results can be further demonstrated by the phosphorescence spectra of these molecules acquired at 77 K in dilute solution, which exhibit similar vibronic emissions as those of the polymers at room temperature

(Supplementary Figs. 32–34). Therefore, compared with previous studies that have achieved polymers with long-lived RTPs by crystal engineering or mix doping, realization of intrinsic polymers with full-color RTPs simply via one-pot preparation is highly significant.

Inspired by the principle of white light three primary colors, herein, ABM-2AMI, PHE-2AMI, and PY-AMI were simultaneously implanted into PAA-NHS via a one-pot reaction (Fig. 3b, Supplementary Fig. 35). A wide range of afterglow ranging from blue to red was achieved by adjusting the chromophore feed ratio (Fig. 3a). In this study, the molar ratio of ABM-2AMI: PAA-NHS was fixed at 1, following with the PY-AMI and PHE-2AMI from 1:8 (TPAP-118) to 8:1 (TPAP-811), the precise structure of the three components was confirmed by [1]H NMR spectra (Supplementary Figs. 11–16). And a blue-to-pink afterglow of ~3 s was noticed after the 365 nm UV excitation source was turned off. Unexpectedly, when the ratio of the three was 5:1:4, the obtained polymer film TPAP-514 demonstrated an attractive white afterglow (Supplementary Movie 7), and the CIE exhibited the color coordinates (0.33, 0.33) (Fig. 3e). Under 365 nm excitation at room temperature, the three-component polymer TPAP-514 demonstrated three emission peaks. The broad peak at 500 nm was attributed to the overlap of the emission peaks of ABM-2AMI and PHE-2AMI, and the emission peaks at 616 and 662 nm were ascribed to PY-AMI. The phosphorescence intensity at 500 nm gradually decreased with an increase in the PY-AMI: ABM-2AMI: PHE-2AMI molar ratio (Fig. 3c). Additionally, the phosphorescence lifetime significantly decreased (Fig. 3d), which may be attributed to the lower $T_1$ of the pyrene group, resulting in a larger singlet-triplet energy gap that is not conducive to the ISCs of the excitons. The low $T_1$ state can facilitate the depletion of triplet excitons via non-radiative pathways. Nevertheless, the introduction of more pyrene groups can also destroy the rigid conformation constructed by slightly crosslinked ABM-2AMI and PHE-2AMI, reduce intramolecular steric hindrance, and thus promote the loss of triplet excitons via the thermal radiation pathway, thereby decreasing the phosphorescence lifetime. Calculation of the non-radiative decay constants of TPAP-712 and TPAP-811 indicated substantially high $k_{nr}^{Phos}$ values of these polymer films (Supplementary Table 3).

Steady-state photoluminescence spectra and two-dimensional excitation–emission spectrum of the TPAP-514 film at room temperature further revealed its white light emission characteristics (Fig. 4a, b). In order to confirm the source of this white light emission, we carried out the following research. First, the temperature-dependent phosphorescence spectra and lifetime changes of all polymers from 80 K to room temperature indicated the characteristics of RTP (Fig. 4c, d, Supplementary Figs. 36, 37), and the interference of thermally activated delayed fluorescence was eliminated. When the ambient temperature was changed from 80 to 280 K, the thermal motions of the molecules intensified, and the triplet excitons were lost in a non-radiative manner. Thus, the phosphorescence intensity gradually decreased, and the phosphorescence lifetime significantly shortened. On this basis, the lifetime of the singlet excited state is much shorter than that of the triplet state, generally at the nanosecond level, while the delayed spectra of all single-component and three-component polymers show long-lived emission, which can well explain that the RTP of TPAP-514 does not come from the singlet contribution of the three component, but from the triplet state.

After that, in these polymer system, the overlap between the absorption bands and emission regions corresponding to the three components is not obvious (Fig. 4f), which does not meet the prerequisite for triplet-triplet energy transfer (TTET), which requires a good overlap integral between the emission spectrum of the donor and the absorption spectrum of the acceptor[48]. In addition, the time-resolved persistent emission spectra of PAPHE, PAABM, PAPY and TPAP-514 were recorded, respectively (Fig. 4e). The persistent phosphorescence of TPAP-514 observed by is obviously from the sum of the emission of the three components, indicating the independent

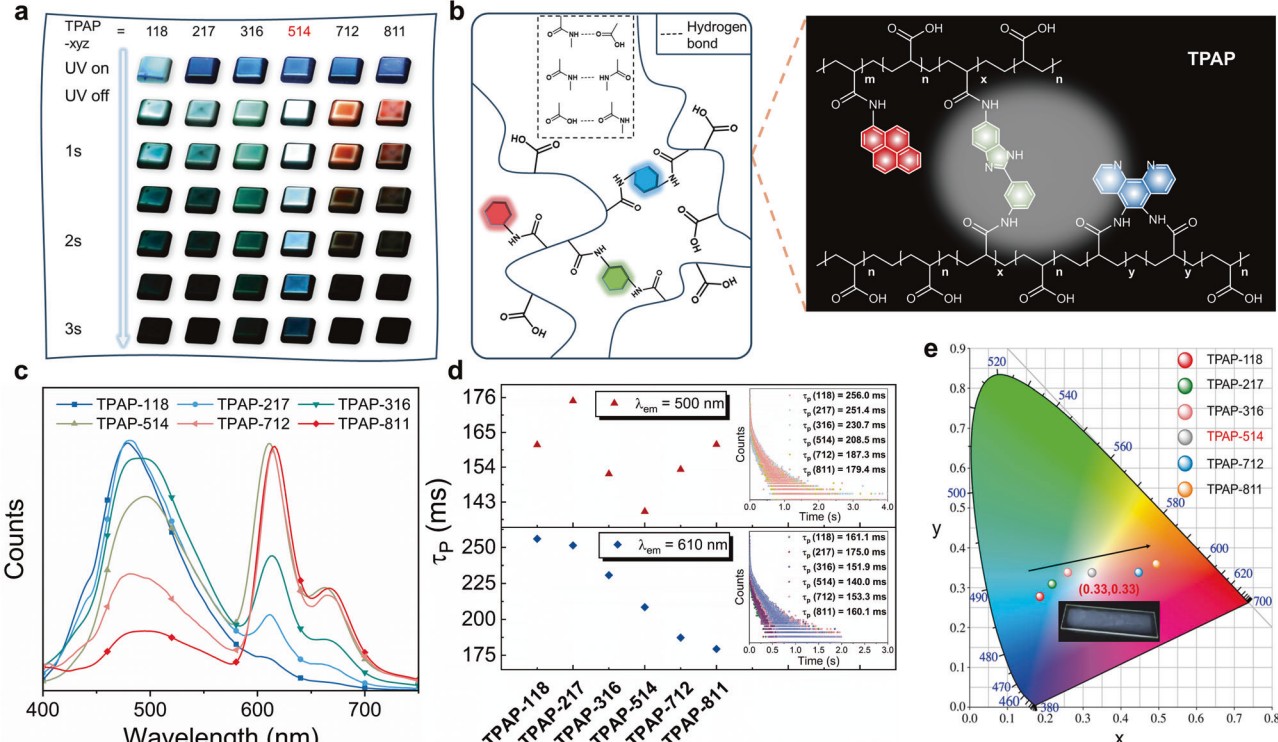

**Fig. 3 | Feasible strategy of three-component regulation of the RTP color of an amorphous polymer. a** Images of the phosphorescences of the films with different proportions of three-component polymer under 365 nm excitation at room temperature. *xyz* represents the PY-AMI: ABM-2AMI: PHE-2AMI ratio. **b** Schematic diagram and local molecular formula of the three-component afterglow-tunable polymer TPAP. The red regular hexagon represents the central group of PY-AMI, the blue represents PHE-2AMI and the green represents ABM-2AMI.

**c** Phosphorescence spectra of three-component polymers with different PY-AMI: ABM-2AMI: PHE-2AMI ratios under 365 nm excitation and environmental conditions. **d** Phosphorescence lifetimes of the three-component polymers with different PY-AMI: ABM-2AMI: PHE-2AMI ratios at 500 nm and 610 nm under environmental conditions ($\lambda_{ex}$ = 365 nm). "118" represents TPAP-118. **e** CIE coordinate diagram corresponding to the phosphorescence spectra of the three-component polymers.

existence of the triplet excitons of the three components. Furthermore, the temperature-dependent steady-state phosphorescence spectrum of TPAP-514 proved the relationship between the three triplet excitons (Fig. 4d). During the temperature from 80 K to 280 K, the emission band intensities at 510 nm and 612 nm decrease synchronously, and the loss rates are 88.4% and 75.7%, respectively. This indicated that there was no obvious Dexter-type TTET channel between the high-level PHE-2AMI and the lower-level ABM-2AMI and PY-AMI[49]. Therefore, the emission of TPAP-514 polymer can also exclude the contribution of TTET. In summary, the long-lived white light of TPAP-514 was proved to be only from the superposition of the triplet excited states of three components.

Naturally, we believe that a phenomenon of response excitation wavelength exists in three-component white light-emitting polymers. When the excitation wavelength is decreased from 365 to 250 nm, the phosphorescence demonstrates a blue shift (Supplementary Fig. 38). From the excitation spectra, we can speculate that the excitation dependence of phosphorescence arises from the different triplet excited states of TPAP-514 (Fig. 4g). As controls, the phosphorescence responses of PAPHE and PAPY at different excitation wavelengths were evaluated, and the results implied that the emission peaks did not significantly change (Supplementary Fig. 39). However, wide-angle X-ray scattering of TPAP-514 only exhibited two broad scattering bands near 1.47 and 2.46 Å corresponding to PAA[50] without considerable peaks originating from π-π interactions of different aggregation states of ABM-2AMI (Supplementary Fig. 40). Therefore, we infer that the excitation wavelength response of the TPAP-514 film may partially arise from the different aggregated forms of ABM-2AMI, and different chromophores demonstrate different responses to excitation

wavelengths. At cryogenic temperatures, the confinement of the slightly crosslinked structure to the molecule is highly intensified, which reduces the heat loss caused by molecular vibration. Therefore, the $T_1$ excitons corresponding to ABM-2AMI and PHE-2AMI are more prone to return to the ground state by radiative transition; thus, the emission intensity at 505 nm is significantly higher than that of the long-wavelength band at 80 K. Variation of the single-component phosphorescence intensity at cryogenic temperatures also supports this point. The fluctuation in the phosphorescence intensity of PAPY induced by the change in the ambient temperature was substantially less than those in the cases of PAPHE and PAABM (Supplementary Fig. 36).

Unexpectedly, after turning off the UV light source for ~2 s, the afterglow color of the TPAP-514 film gradually changed from white to blue (Fig. 3d). Additionally, a similar phenomenon was observed in other three-component polymer films, revealing time-dependent phosphorescences of these films. The time-resolved spectra of TPAP-514 film under 365 nm excitation can well explain this phenomenon. The film exhibited a long lifetime of 197.2 ms at 494 nm and a short lifetime of 140.0 ms at 610 nm (Supplementary Fig. 41). Importantly, the intensity of the 610 nm emission band rapidly decreased with an increase in the delay time from 2 to 100 ms (Fig. 4h). Thus, the calculated decay rates of the PHE-2AMI excitons were considerably slower than those of the PY-AMI excitons. This result indicates that discoloration originates from the different decay rates of the triplet excitons of the three introduced phosphorescence units.

To understand the potential mechanism of colorful long-lived RTPs, grafting of the phosphorescent units onto the polymers was simulated, and the phosphorescent units grafted onto the polymers

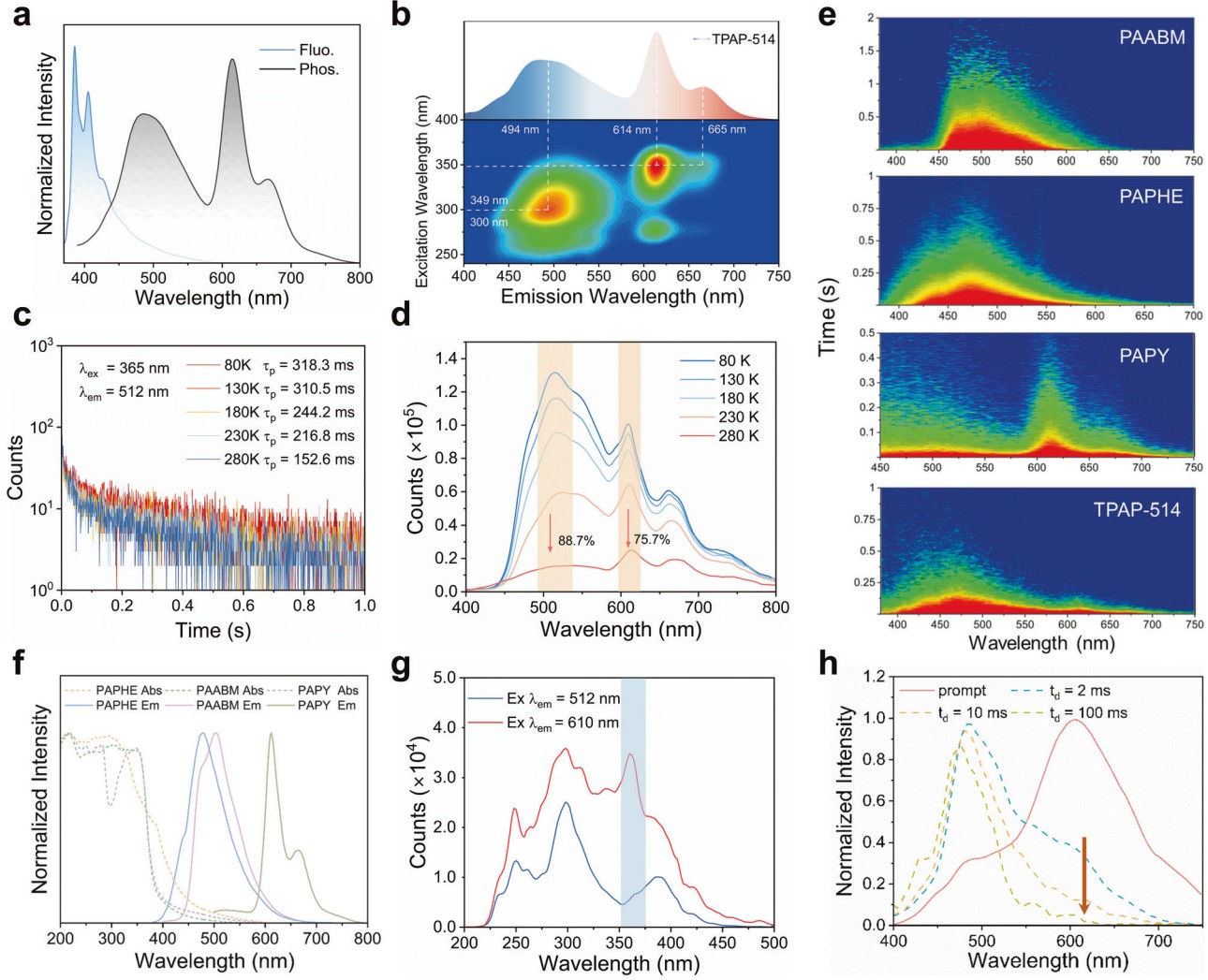

**Fig. 4 | Luminescence properties of the white light-emitting TPAP-514 film.**
**a** Steady-state photoluminescence spectra of the TPAP-514 film excited at 365 nm.
**b** Two-dimensional excitation–emission spectra of TPAP-514 film (excitation wavelength from 200 to 400 nm, emission wavelength from 400 to 750 nm). **c** The phosphorescence decay curves of 512 nm emission band of TPAP-514 film under 365 nm excitation in the temperature range of 80–280 K. **d** Temperature-dependent phosphorescence spectra of TPAP-514 films from 80 to 280 K ($\lambda_{ex}$ = 365 nm). **e** Time-resolved phosphorescence emission spectra of PAABM, PAPHE, PAPY, TPAP-514 at room temperature under 365 nm excitation. **f** Absorption and emission spectra of three kinds of one-component polymer films at room temperature. **g** Excitation spectra of TPAP-514 film obtained at 512 and 610 nm under room temperature, respectively. **h** Emission spectra of prompt and delayed phosphorescence of TPAP-514 at different delay times ($t_d$ = 2–100 ms).

were analyzed by density functional theory (DFT) calculations. The highest occupied molecular orbital-lowest unoccupied molecular orbital energy gaps of PAA-NHS (4.90 eV) were sufficiently large to cover the gaps of PAPHE, PAABM, PAACR, PAPY, and TPAP, thereby avoiding electron transfer between the polymer main chains and phosphors (Supplementary Fig. 42). Electrostatic potential (ESP) on the carboxyl group of the main chain surface of TPAP was positive (Fig. 5a), which was as high as up to 53.16 kcal mol$^{-1}$, and all the organic phosphors demonstrated negative ESPs on their backbones; particularly, ESP of the N atomic part was as low as -47.34 kcal mol$^{-1}$. This large ESP difference enables the formation of strong electrostatic interactions between the phosphor and polymer backbone, thereby inhibiting the non-radiative transitions of triplet excitons. Furthermore, the SOC constants, which are very important for ISCs, were calculated via time-dependent DFT. All the polymers exhibited abundant ISC channels between $S_1$ and $T_n$ with large constants, which were attributed to the presence of heteroatoms (N and O) and aromatic skeletons in these polymers (Fig. 5b). For instance, the electronic configuration verified by the analysis of the natural transition orbitals of $S_1$ in TPAP

demonstrates a mixture of n-π* and π-π* transitions; in contrast, $T_1$ exhibits a pure π-π* transition, and its $^3$(π-π*) configuration can demonstrate a long-lived emission due to the forbidden transition (Fig. 5c). Similarly, the identical excited triplet energy levels and SOC matrices of the other four polymer units are also depicted in Supplementary Figs. 43–46 to identify the possible ISC channels. Therefore, the experimental results and theoretical calculations further reveal that white-light emission of TPAP originates from the superposition of the triplet states of different structural units (Fig. 5d).

After revealing the long-lived RTP mechanism of amorphous polymers, we explored the potential applications of the prepared materials in the field of light-emitting diodes. The single-component and three-component polymers were attached to the surface of UV LED ($\lambda_{ex}$ = 365 nm) in the form of films by spin coating to fabricate a film lampshade (Fig. 6a). The LED assembled with the film lampshade and unassembled LED together formed an array that emitted bright luminescence (Fig. 6b). After the power supply was cut off, the LEDs applied to the film lampshade exhibited attractive multicolor afterglow, which were demonstrated as "C", "Q", "U", and "T" patterns.

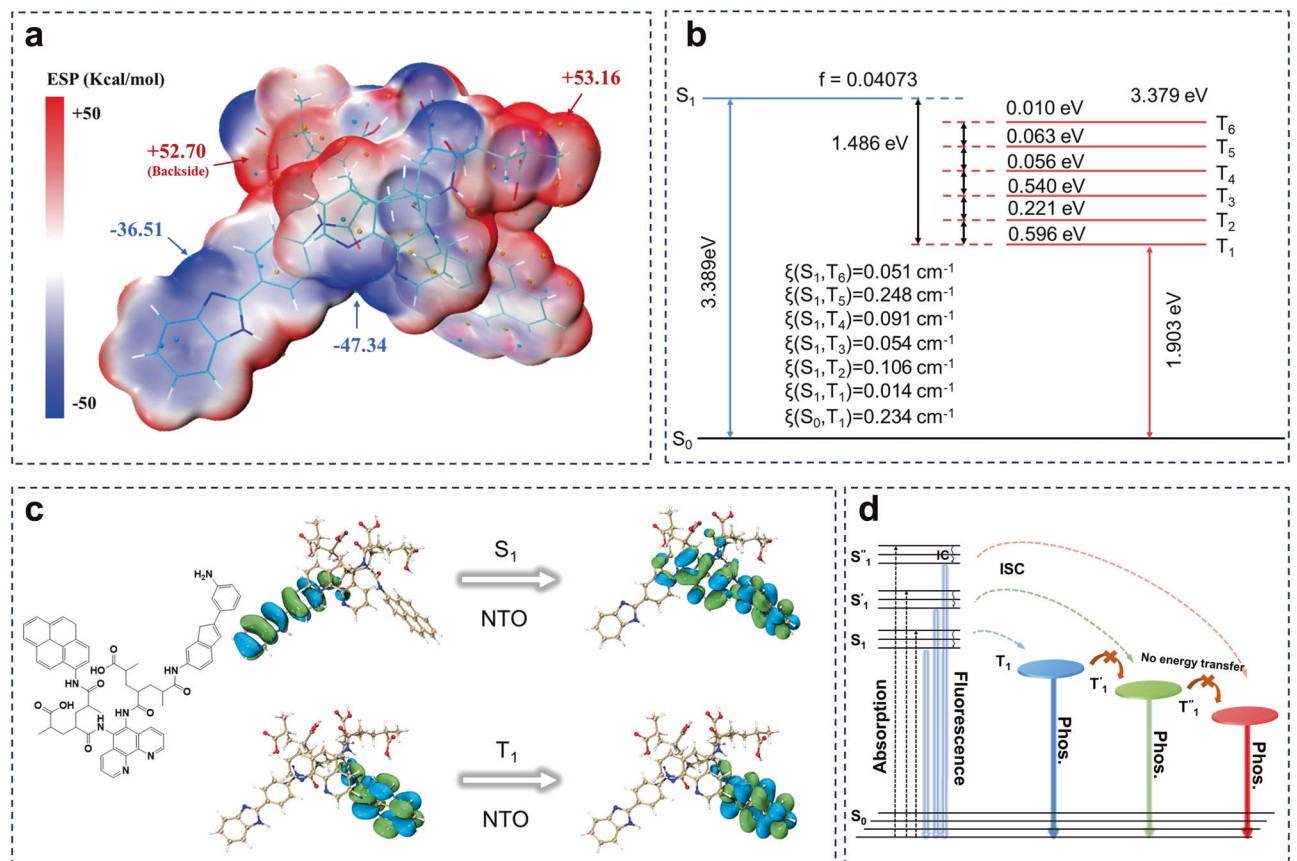

**Fig. 5 | Theoretical calculation of luminescent polymer structural units and white light-emission mechanism. a** Electrostatic potential (ESP) of afterglow-tunable TPAP determined by simulation. **b** Energy levels and spin-orbit coupling constants of TPAP. **c** Natural transition orbitals of singlet and triplet excited states of TPAP. **d** White light-emission mechanism of different excited states. "Phos." stands for phosphorescence. "IC" represents the internal conversion based on Kasha rule.

Interestingly, by programmatically regulating the external circuit of the LED array, the afterglow exhibited different display paths. Additionally, the variable driving voltage endowed the three-component white afterglow film LED assemblies with different emission efficiencies (Fig. 6c). Because of its excitation and emission characteristics, PAACR was applied to dynamic anti-counterfeiting. PAACR solution was solidified in a mold to obtain the dry letter pattern "PRTP" (Fig. 6d). Under visible-light excitation, similar to the case of exposure to natural light, the pattern demonstrated the original color of the polymer; in contrast, under 365 nm UV excitation, the pattern exhibited bright blue fluorescence (Fig. 6e), obtaining additional information based on fluorescence. After removing the two excitation sources, both patterns demonstrated yellow RTP signals with different intensities.

## Discussion

In summary, we developed a simple strategy to covalently couple four amino-containing conjugated chromophores separately with a copolymer precursor (PAA-NHS) by a one-pot reaction. The resulting polymer films PAABM, PAPHE, PAACR, and PAPY emitted long-lived RTPs with different colors under environmental conditions. Owing to the introduction of the heteroatom N, aromatic structures, and dense distributions of carbonyl groups, these films demonstrated high SOC constants and abundant ISC channels, which also contributed to the construction of H bond networks. Note that PAABM and PAPHE exhibited longer phosphorescence lifetime of 334.7 ms and the highest phosphorescence quantum yield of 14.7%, respectively. More importantly, white RTP of the intrinsic polymer TPAP under environmental conditions was realized when the ratio of the three was 5:1:4.

Combined results of comparisons, photophysical analysis, and theoretical calculations suggest that white RTP originates from different triplet excited states. This study provides a new strategy for the convenient and efficient preparation of intrinsic polymers with diverse long-lived RTPs and a wide range of applications in various fields such as LEDs.

## Methods

### Materials

Unless otherwise stated, the reagents used in the experiment including acrylic acid (AA, 98%), acrylic acid-N-succinimide ester (AA-NHS, 98%), 2-(3-aminophenyl)-5-aminobenzimidazole (ABM-2AMI, 95%), 5,6-diamino-1,10-phenanthroline (PHE-2AMI, 98%), 9-aminoacridine (ACR-AMI, 97%), 1-aminopyrene (PY-AMI, 97%) were purchased from energy chemical plants without further purification. Deionized water was prepared in the laboratory. Ethyl acetate (EA, analytically pure), dimethyl sulfoxide (DMSO, analytically pure), purchased from Chengdu Cologne Chemical Co., Ltd. DMSO was dried by molecular sieve, and the remaining solvent was not further purified. Azobisisobutyronitrile (AIBN, 98%) was purchased from an energy chemical plant and used after recrystallization.

### Synthesis of PAACR-15

AA-NHS (0.17 g, 0.001 mol) was fully dissolved in DMSO by ultrasonication. The resulting colorless transparent solution was added to a 100 mL polymerization tube along with AA (5.04 mL, 0.07 mol) and AIBN (0.057 g, 1 wt% of the total mass of the monomer). Thereafter, 20 mL DMSO was introduced into the reaction mixture for vacuum degassing and argon (Ar) circulation for 5 times, and the reaction was

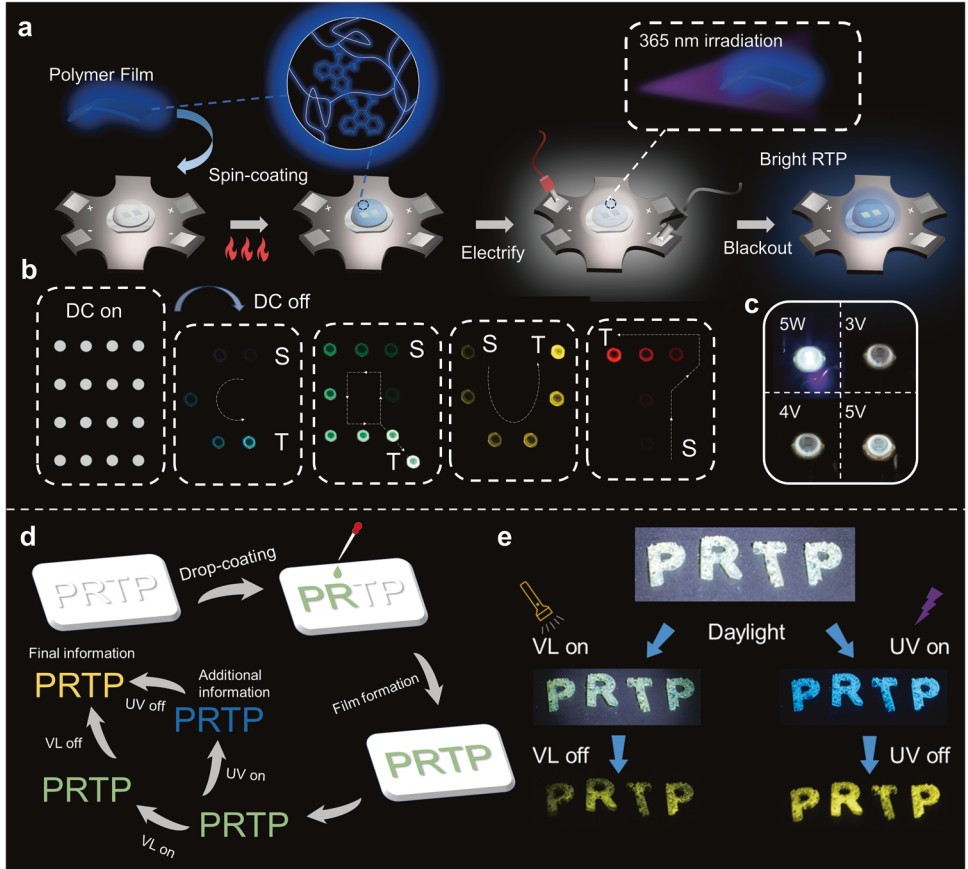

**Fig. 6 | Application of polymeric RTP materials in light-emitting diodes (LEDs) and dynamic anti-counterfeiting. a** Schematic of the synthesis of LED afterglow lampshade. **b** One component polymeric RTP material for LED display. DC on and off represent the switching on and off of power supply, respectively. The arrow indicates the direction of the afterglow path. "S" stands for the starting point and "T" stands for the terminal point. **c** Images of three-component white light emitter for afterglow lampshade under different driving voltages. **d** Processing schematic of PAACR for dynamic anti-counterfeiting. **e** Luminescence images of the one-component polymer model material PAACR under natural light, white light flash excitation, and 365 nm ultraviolet light excitation. "VL" (electric torch) denotes visible light.

conducted at 65 °C for 48 h. After the reaction, a pale yellow solution with a certain viscosity was obtained, which was transferred to a 100 mL single-mouth flask without treatment. Then, ACR-AMI (0.029 g, 0.15 mmol) was added to the abovementioned solution followed by the introduction of a small amount of solvent to reduce the viscosity of the system; subsequently, the reaction was performed overnight at room temperature. After completion of the reaction, PAACR-15 was poured into ethyl acetate for precipitation and then used after three cycles of treatment.

$^1$H NMR (400 MHz, DMSO-$d_6$, ppm): δ 12.55 (s, 11H), 8.61 (d, J = 8.6 Hz, 2H), 8.01-7.93 (m, 2H), 7.90-7.84 (m, 1H), 7.57 (ddd, J = 8.1, 6.8, 1.3 Hz, 2H), 4.15 (s, 1H), 2.71 (s, 1H), 2.54 (s, 56H), 2.37 (s, 2H), 2.22 (s, 12H), 1.75 (s, 2H), and 1.46 (s, 3H). FTIR (cm$^{-1}$): 3502(vw), 2931(w), 2548(w), 1939(w), 1712(s), 1444(w), 1404(w), 1317(vw), 1242(w), 1164(m), 1108(s), 1006(s), 943(vw), 800(w), and 709(w).

**Synthesis of PAABM**

AA-NHS (0.17 g, 0.001 mol) was fully dissolved in DMSO by ultrasonication. The resulting colorless transparent solution was added to a 100 mL polymerization tube along with AA (5.04 mL, 0.07 mol) and AIBN (0.057 g, 1 wt% of the total mass of the monomer). Thereafter, 20 mL DMSO was introduced into the reaction mixture for vacuum degassing and argon (Ar) circulation for 5 times, and the reaction was conducted at 65 °C for 48 h. After the reaction, a pale yellow solution with a certain viscosity was acquired, which was then transferred to a 100 mL single-mouth flask without treatment. Subsequently, ABM-

2AMI (0.020 g, 0.09 mmol) was added to the abovementioned solution followed by the introduction of a small amount of solvent to reduce the viscosity of the system; thereafter, the reaction was performed overnight at room temperature. After completion of the reaction, PAABM was poured into ethyl acetate for precipitation and then used after three cycles of treatment.

$^1$H NMR (400 MHz, DMSO-$d_6$, ppm): δ 12.17 (s, 5H), 7.64-6.90 (m, 1H), 6.79-6.16 (m, 0H), 4.15 (s, 0H), 2.71 (s, 0H), 2.59 (s, 0H), 2.54 (s, 28H), 2.21 (s, 2H), 1.75 (s, 1H), and 1.51 (s, 2H). FTIR (cm$^{-1}$): 3537(vw), 2922(w), 2536(w), 1953(w), 1712(s), 1608(m), 1446(m), 1315(vw), 1244(w), 1157(m), 1110(s), 1006(s), 937(vw), 802(w), and 715(w).

**Synthesis of PAPHE**

AA-NHS (0.17 g, 0.001 mol) was fully dissolved in DMSO by ultrasonication. The resulting colorless transparent solution was added to a 100 mL polymerization tube along with AA (5.04 mL, 0.07 mol) and AIBN (0.057 g, 1 wt% of the total mass of the monomer). Thereafter, 20 mL DMSO was introduced into the reaction mixture for vacuum degassing and argon (Ar) circulation for 5 times, and the reaction was conducted at 65 °C for 48 h. After the reaction, a pale yellow solution with a certain viscosity was obtained, which was then transferred to a 100 mL single-mouth flask without treatment; subsequently, PHE-2AMI (0.019 g, 0.09 mmol) was added to the abovementioned solution followed by the introduction of a small amount of solvent to reduce the viscosity of the system; then, the reaction was performed overnight at room temperature. After completion of the reaction, PAPHE was

poured into ethyl acetate for precipitation and then used after three cycles of treatment.

$^1$H NMR (400 MHz, DMSO-$d_6$, ppm): δ 12.24 (s, 1H), 8.43-7.29 (m, 1H), 4.21 (d, J = 48.7 Hz, 0H), 2.71 (s, 0H), 2.54 (s, 6H), and 2.20 (s, 1H). FTIR (cm$^{-1}$): 3510(vw), 2924(w), 2549(w), 1950(w), 1708(s), 1438(w), 1408(w), 1317(vw), 1246(w), 1163(m), 1110(s), 1002(s), 943(vw), 800(w), and 709(w).

### Synthesis of PAPY

AA-NHS (0.17 g, 0.001 mol) was fully dissolved in DMSO by ultra-sonication. The resulting colorless transparent solution was added to a 100 mL polymerization tube along with AA (5.04 mL, 0.07 mol) and AIBN (0.057 g, 1 wt% of the total mass of the monomer). Subsequently, 20 mL DMSO was introduced into the reaction mixture for vacuum degassing and argon (Ar) circulation for 5 times, and the reaction was conducted at 65 °C for 48 h. After the reaction, a pale yellow solution with a certain viscosity was acquired, which was then transferred to a 100 mL single-mouth flask without treatment; thereafter, PY-AMI (0.019 g, 0.09 mmol) was added to the abovementioned solution followed by the introduction of a small amount of solvent to reduce the viscosity of the system; subsequently, the reaction was performed overnight at room temperature. After completion of the reaction, PAPY was poured into ethyl acetate for precipitation and then used after three cycles of treatment.

$^1$H NMR (400 MHz, DMSO-$d_6$, ppm): δ 12.22 (s, 0H), 8.61-7.24 (m, 0H), 4.15 (s, 0H), 2.65 (d, J = 47.0 Hz, 0H), 2.54 (s, 1H), 2.37 (s, 0H), 2.31 (d, J = 14.1 Hz, 0H), 2.21 (s, 0H), and 2.36–1.96 (m, 0H). FTIR (cm$^{-1}$): 3527(vw), 2929(w), 2549(w), 1946(w), 1707(s), 1446(w), 1406(w), 1319(vw), 1244(w), 1165(m), 1109(s), 999(s), 941(vw), 800(w), and 707(w).

### Synthesis of TPAP-514 (mol. PY-AMI:ABM-2AMI:PHE-2AMI = 5:1:4)

AA-NHS (0.17 g, 0.001 mol) was fully dissolved in DMSO by ultra-sonication. The resulting colorless transparent solution was added to a 100 mL polymerization tube along with AA (5.04 mL, 0.07 mol) and AIBN (0.057 g, 1 wt% of the total mass of the monomer). Thereafter, 20 mL DMSO was introduced into the reaction mixture for vacuum degassing and argon (Ar) circulation for 5 times, and the reaction was conducted at 65 °C for 48 h. After the reaction, a pale yellow solution with a certain viscosity was obtained, which was then transferred to a 100 mL single-mouth flask without treatment; subsequently, PY-AMI (9.8 mg, 0.045 mmol), PHE-2AMI (7.6 mg, 0.036 mmol), and ABM-2AMI (2.1 mg, 0.009 mmol) were simultaneously added to the abovementioned solution followed by the introduction of a small amount of solvent to reduce the viscosity of the system; thereafter, the reaction was performed overnight at room temperature. After completion of the reaction, TPAP-514 was poured into ethyl acetate for precipitation and then used after three cycles of treatment.

$^1$H NMR (400 MHz, DMSO-$d_6$, ppm): δ 12.24 (s, 0H), 8.25 (d, J = 9.4 Hz, 0H), 8.18-7.95 (m, 0H), 8.02-7.83 (m, 0H), 7.94-7.65 (m, 0H), 7.37 (s, 0H), 7.35 (s, 0H), 4.22 (d, J = 47.9 Hz, 0H), 2.59 (s, 0H), 2.54 (s, 2H), and 2.21 (s, 0H). FTIR (cm$^{-1}$): 3522(vw), 2927(w), 2543(w), 1945(w), 1709(s), 1446(w), 1408(w), 1319(vw), 1245(w), 1168(m), 1104(s), 997(s), 943(vw), 802(w), and 707(w).

The polymerization products TPAP-118, TPAP-217, TPAP-316, TPAP-712, and TPAP-811 were obtained under the same conditions as TPAP-514 except for the feed proportion of the small molecules. The molar ratio of precursor AA: AA-NHS was fixed at 70: 1.

### TPAP-118 (mol. PY-AMI:ABM-2AMI:PHE-2AMI = 1:1:8)

PY-AMI (2.0 mg, 0.009 mmol), ABM-2AMI (2.1 mg, 0.009 mmol), PHE-2AMI (15.2 mg, 0.072 mmol).

### TPAP-217 (mol. PY-AMI:ABM-2AMI:PHE-2AMI = 2:1:7)

PY-AMI (3.9 mg, 0.018 mmol), ABM-2AMI (2.1 mg, 0.009 mmol), PHE-2AMI (13.3 mg, 0.063 mmol).

### TPAP-316 (mol. PY-AMI:ABM-2AMI:PHE-2AMI = 3:1:6)

PY-AMI (5.9 mg, 0.027 mmol), ABM-2AMI (2.1 mg, 0.009 mmol), PHE-2AMI (11.4 mg, 0.054 mmol).

### TPAP-712 (mol. PY-AMI:ABM-2AMI:PHE-2AMI = 7:1:2)

PY-AMI (13.7 mg, 0.063 mmol), ABM-2AMI (2.1 mg, 0.009 mmol), PHE-2AMI (3.8 mg, 0.018 mmol).

### TPAP-811 (mol. PY-AMI:ABM-2AMI:PHE-2AMI = 8:1:1)

PY-AMI (15.7 mg, 0.072 mmol), ABM-2AMI (2.1 mg, 0.009 mmol), PHE-2AMI (1.9 mg, 0.009 mmol).

### Film fabrication

The fully purified polymer solution was placed in a centrifuge tube, and then, a small amount of DMSO was added to dilute it; thereafter, ultrasonication was used to completely disperse the resulting solution. Subsequently, the resulting solution was placed dropwise on a 10 × 10 × 1 mm quartz sheet followed by drying in a vacuum oven at 90 °C for 3 h to obtain luminescent polymer film.

### Measurements

The samples were measured at the solution and film state. Fourier-transform infrared (FT-IR) spectra were measured using a PerkinElmer Spectrum Two Infrared Spectrometer using the direct transmission technique. Nuclear magnetic resonance ($^1$H NMR, $^{13}$C NMR) spectra were recorded at a DRX-400 and 101 MHz (Bruker) superconducting-magnet NMR spectrometer using deuterated dimethyl sulfoxide and deuterated chloroform as the solvent. Molecular weights of the polymers were measured by gel permeation chromatography (GPC) on a TRSEC MODEL302 using DMSO as the eluent and polystyrene as the standard. Powder X-ray diffraction (PXRD) measurements were recorded on the Rigaku D/Max-2500 X-ray diffractometer at room temperature, using Cu Kα radiation with 2θ range of 2–70°, 40 KeV, and 30 mA. The scanning rate was 0.01° s$^{-1}$ (2θ). Differential scanning calorimetry (DSC) was carried out in nitrogen atmosphere on TA Q20 at a heating rate of 20 °C min$^{-1}$. UV-Vis absorption spectra were conducted on a Hitachi UH 5300 UV-Vis Spectrophotometer. Photophysical properties were characterized using Edinburgh FLS1000 fluorescence spectrophotometer and Hitachi F-4700 fluorescence spectrophotometer, and the samples were in the thin-film state. Phosphorescence spectra were performed using microsecond flash lamp with frequency of 100 Hz and delay time of 0.5 ms. Fluorescence spectra were recorded with a xenon lamp. The phosphorescence lifetime was measured by the dynamic lifetime measurement mode with a xenon lamp. The time-resolved emission spectra were recorded on FLS1000 fluorescence spectrophotometer with microsecond flash lamp having frequency of 1 Hz. Phosphorescence quantum yield was collected on Edinburgh FLS1000 fluorescence spectrophotometer equipped with an integrating sphere by "direct excitation" method. The photos were taken by a Canon EOS 80D camera in environmental conditions.

### Theoretical calculation method

Geometry optimization the phosphorescent units grafted in polymers PAPHE, PAABM, PAACR, PAPY, and TPAP were realized by DFT with dispersion-corrected theory (D3) at the B3LYP-D3/def2-SVP level[51]. And their excitation energies and Spin-orbit coupling (SOC) constants were calculated at the PBE0/def2-SVP level with TD-DFT method[52]. The ESP map and values of TPAP were calculated using B3LYP-D3 with ma-def2-TZVP basis set. ORCA 5.0.4 program package was employed for all

quantum chemistry calculations[53]. The electron transition characterization and natural transition orbitals were obtained by electron excitation analysis performed using the Multiwfn 3.8 (dev) code[54]. The isosurface maps of various orbitals and ESP maps were rendered by means of Visual Molecular Dynamics (VMD)[55] software based on the files exported by Multiwfn[56].

## Data availability

All data supporting the findings of this study are available within the article and its Supplementary Information files, as well as from the corresponding author upon request.

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

## Acknowledgements

This work was financially supported by the NSFC (22275025, 52003032), Natural Science Foundation of Chongqing (cstc2020jcyj-msxmX0556), Science and Technology research program of Chongqing Municipal Education Commission (KJQN202001104), Innovation Research Group at Institutions of Higher Education in Chongqing (CXQT19027), the Chongqing Talent Program (CQYC2020057922), the Science and Technology Project of Banan District, the Innovation Support Plan for the Returned Overseas of Chongqing (cx2020052), and the Open Fund of Guangdong Provincial Key Laboratory of Luminescence from Molecular Aggregates (2021-kllma-03) and Postgraduate Research and Innovation Program in Chongqing (gzlcx20232002).

## Author contributions

Q.A.C. and C.L.Y. conceived and were responsible for the experiments. Q.A.C., H.H., Y.K.W., C.L. and Y.Z. synthesized the polymers and performed the photophysical data measurements. X.H.C., Q.Z. and Y.Y. completed the preparation of the application. L.J.Q. performed theoretical calculations. J.Y.H. carried out data collation. Q.A.C., L.J.Q. and C.L.Y. performed the data analysis and wrote the manuscript. All authors contributed to the final version of the manuscript.

## Competing interests

The authors declare no competing interests.
