## [Peer Review File · Nature Communications]

REVIEWER COMMENTS

Reviewer #1 (Remarks to the Author):

In this work, Prof. Yang and coworkers used a strategy of covalently coupling different conjugated chromophores with poly(acrylic acid (AA)-AA-N-succinimide ester) (PAA-NHS) by a simple and rapid one-pot reaction to obtain polymers with long-lived RTPs of various colors (the afterglow colors of polymers can be modulated from blue to red). Among these polymers, the highest phosphorescence quantum yield of PAPHE reaches 14.7%. Polymer TPAP-514 exhibited a white afterglow at room temperature with the chromaticity coordinates (0.33, 0.33) when the ratio of chromophores reaches a suitable value owing to the three-primary-color mechanism. However, such white emission would be very hard to be contributed only from triplet states? The authors should provide more convincing experimental data to confirm the emissions discussed in the text resulting from ONLY triplet states (without singlet states contribution from three components or excluding triplet-triplet energy transfer processes). The precise structures shown for multi-components copolymers should be confirmed by convincing experimental data. The one-pot reaction for multi-components copolymerization would NOT be 100% to obtain pre-ratios copolymers. The materials obtained in this work were actually mixtures, so 'intrinsic' RTP might mislead readers. The authors should provide data about reproducibility. In addition, so-called 'applications of the obtained polymers in light-emitting diodes' shown in Fig. 6 were still photo-excited RTP polymers films and were NOT real 'OLED' devices with electric-driven (such RTP polymers acted as active layer). In fact, polymeric room-temperature phosphorescence (RTP) materials had been reported by many scientists (Prof. Z. Li with many works e.g. Refs. [1, 19], Prof. Y-L. Zhao with many works e.g. Ref.[7], Prof. Marder with Ref.[10], Prof. Ogoshi with Ref.[22], Prof. Z. An with many works e.g. Ref.[30], Prof. X. Ma with many works e.g. Recent Advances of Pure Organic Room Temperature Phosphorescence Based on Functional Polymers. Accounts Materials Research, 2023, 10.1021/accountsmr.3c00090). Therefore, this work lacks novelty and could NOT be accepted for publication in Nature Comm.

Reviewer #2 (Remarks to the Author):

In this work, Yang and co-authors present a strategy of covalently coupling different conjugated chromophores with poly(acrylic acid (AA)-AA-N-succinimide ester) by a simple and rapid one-pot reaction to obtain slightly crosslinked polymers with long-lived RTP of various colors. Interestingly, the afterglow colors of polymers can be modulated from blue to red by introducing three chromophores into them, the acquired polymer TPAP-514 exhibited pure white afterglow at room temperature with the chromaticity coordinates (0.33, 0.33). Moreover, the potential applications of the obtained polymers in light-emitting diodes and dynamic anti-counterfeiting are explored. I think the proposed strategy provides a new idea for constructing intrinsic polymers with diverse RTPs. Based on the rational material

preparation and the outstanding optical performance, it can be recommended for publication in Nature Communications after minor revision.

1. In this work, only ^1H NMR and FTIR were used to characterize the polymers. ^{13}C NMR spectra of PAPHE, PAABM, PAACR, PAPY is recommended to be used for further demonstration of their chemical structures.
2. How did you confirm the long-lived luminescence of polymers from RTP rather TADF?
3. For polymer PAPHE, how did you separate fluorescence and phosphorescence spectra when calculated the quantum yield?
4. The Figures 2 and 3 should be clearly displayed, some font size for Figure 2 should be increased.
5. The GPC data of PAABM, PAACR, PAPY should be provided.
6. In Fig. S13, S14, the TGA and DSC curves of all polymers PAPHE, PAABM, PAACR, PAPY should be added.

Reviewer #3 (Remarks to the Author):

In this work, the author reported a one-pot reaction to covalently introduce conjugated chromophores into poly(acrylic acid (AA)-AA-N-succinimide ester) (PAA-NHS) to construct a micro-crosslinked long-lived intrinsic polymer RTP system. The intrinsic polymer PAPHE obtained by their strategy has a phosphorescence quantum yield of 14.7 %. Importantly, the polymer film achieves a wide range of color-tunable afterglow from blue to red including white light by rationally adjusting the feed ratio of the phosphorescent unit through multiple components. The multi-color long-life luminescence properties based on these polymers have also been reasonably applied to LED and anti-counterfeiting fields. This work has a great contribution to the development of polymer RTP field, especially provides a novel strategy for the development of white light RTP system. I think this manuscript will arouse the widely interest of readers of Nature Communications, so it can be recommended to publish after minor revision.

1. To further confirm the chemical structure of desired polymer, ^{13}C NMR data should be added in revised manuscript.
2. In Fig.2h, the authors mentioned that the micro-crosslinked PAABM and PAPHE polymer films stimulated the construction of a rigid environment, but why PAACR exhibited longer phosphorescence (395.9 ms)?
3. In Figure 3d, due to the existence of multiple emission centers, the multi-component polymer should exhibit different phosphorescence lifetimes in different emission bands. Therefore, the author should

confirm whether the collected phosphorescence lifetime comes from the same emission band, because there is no annotation here.

4. In the application as a afterglow LED lampshade, the picture is not particularly clear, and in Figure 6b, what is the meaning of "S", "T"?

5. The color of annotation of figures S16, S17 and S20 should be black.

Reviewer #1 (Remarks to the Author):

In this work, Prof. Yang and coworkers used a strategy of covalently coupling different conjugated chromophores with poly(acrylic acid (AA)-AA-N-succinimide ester) (PAA-NHS) by a simple and rapid one-pot reaction to obtain polymers with long-lived RTPs of various colors (the afterglow colors of polymers can be modulated from blue to red). Among these polymers, the highest phosphorescence quantum yield of PAPHE reaches 14.7%. Polymer TPAP-514 exhibited a white afterglow at room temperature with the chromaticity coordinates (0.33, 0.33) when the ratio of chromophores reaches a suitable value owing to the three-primary-color mechanism.

Response: We appreciate the valuable comments from the reviewer on our work. As can be seen, the comments are vital improvement of the quality of the revised manuscript. Our point-to-point response is presented below. We hope that our responses, along with optimized details, can clear the issues raised by the reviewer.

1. However, such white emission would be very hard to be contributed only from triplet states? The authors should provide more convincing experimental data to confirm the emissions discussed in the text resulting from ONLY triplet states (without singlet states contribution from three components or excluding triplet-triplet energy transfer processes).

Response: We are quite grateful to you for pointing out this question, and apologize for not clearly enough excluding the singlet contribution of the three components and the triplet-triplet energy transfer process. First, we have supplemented the temperature-dependent phosphorescence spectra and decay curves of all single-component polymers to exclude the effect of thermally activated delayed fluorescence (Figures R1 and R2) (Supplementary Figures 34 and 35). On this basis, the lifetime of the singlet excited state is much shorter than that of the triplet state, generally at the nanosecond level, while the delayed spectra of all single-component and three-component polymers show long-lived emission (> 100 ms), which can well explain that the RTP of the three-

component polymer does not come from the singlet contribution of the three component, but from the triplet state.

Figure R1 (Supplementary Figure 34). a-d Phosphorescence spectra of PAPHE (a), PAABM (b), PAACR (c) and PAPY (d) films excited at 365 nm in the temperature range of 80-280 K.

Figure R2 (Supplementary Figure 35). a-d Phosphorescence decay curves of PAPHE (a), PAABM (b), PAACR (c) and PAPY (d) films excited at 365 nm in the temperature range of 80-280 K.

After that, for triplet-triplet energy transfer (TTET), based on Dexter theory, it is necessary to have a good overlap integral between the emission spectrum of the donor and the absorption spectrum of the acceptor. In our polymer system, the overlap between the absorption band and the emission region corresponding to the three components is not obvious (Figure R3). In addition, the time-resolved persistent emission spectra of PAPHE, PAABM, PAPY and TPAP-514 were recorded respectively. The persistent phosphorescence of TPAP-514 observed is obviously from the sum of the emission of the three components (Figure R4), indicating the independent existence of the triplet excitons of the three components. Furthermore, the temperature-dependent steady-

state phosphorescence spectrum of TPAP-514 proves the relationship between the three triplet excitons. During the temperature from 80 K to 280 K, the emission band intensities at 510 nm and 612 nm decrease synchronously, and the loss rates are 88.7 % and 75.7 %, respectively (Figure R5). This indicates that there is no obvious Dexter-type TTET channel between the high-level PHE-2AMI and the lower-level ABM-2AMI and PY-AMI. In summary, the emission of the three-component polymer can also exclude the contribution of TTET.

Figure R3. a) Absorption and emission spectra of three kinds of one-component polymer films at room temperature.

Figure R4. Time-resolved phosphorescence emission spectra of PAPHE, PAABM, PAPY, TPAP-514 films under 365 nm excitation at room temperature.

Figure R5. a Temperature-dependent phosphorescence spectra of TPAP-514 films from 80 to 280 K ($\lambda_{\text{ex}} = 365$ nm). **b** Temperature-dependent intensity of emission bands at 510 and 612 nm.

According to your suggestion, we have supplemented the relevant part of the description in the revised version, remade Figure 4 and modified Figure 5d. On page 15, we emended the text as follows:

“Steady-state photoluminescence spectra and two-dimensional excitation–emission spectrum of the TPAP-514 film at room temperature further revealed its white light emission characteristics (Figs. 4a and b). In order to confirm the source of this white light emission, we carried out the following research. First, the temperature-dependent phosphorescence spectra and lifetime changes of all polymers from 80 K to room temperature indicated the characteristics of RTP (Figs. 4c and d, Supplementary Figures 34 and 35), and the interference of thermally activated delayed fluorescence was eliminated. When the ambient temperature was changed from 80 to 280 K, the thermal motions of the molecules intensified, and the triplet excitons were lost in a non-radiative manner. Thus, the phosphorescence intensity gradually decreased, and the phosphorescence lifetime significantly shortened. On this basis, the lifetime of the singlet excited state is much shorter than that of the triplet state, generally at the nanosecond level, while the delayed spectra of all single-component and three-component polymers show long-lived emission, which can well

explain that the RTP of TPAP-514 does not come from the singlet contribution of the three component, but from the triplet state.

After that, in these polymer system, the overlap between the absorption bands and emission regions corresponding to the three components is not obvious (Fig. 4f), which does not meet the prerequisite for triplet-triplet energy transfer (TTET), which requires a good overlap integral between the emission spectrum of the donor and the absorption spectrum of the acceptor⁴⁹. In addition, the time-resolved persistent emission spectra of PAPHE, PAABM, PAPY and TPAP-514 were recorded, respectively (Fig. 4e). The persistent phosphorescence of TPAP-514 observed by is obviously from the sum of the emission of the three components, indicating the independent existence of the triplet excitons of the three components. Furthermore, the temperature-dependent steady-state phosphorescence spectrum of TPAP-514 proved the relationship between the three triplet excitons (Fig. 4d). During the temperature from 80 K to 280 K, the emission band intensities at 510 nm and 612 nm decrease synchronously, and the loss rates are 88.4 % and 75.7 %, respectively. This indicated that there was no obvious Dexter-type TTET channel between the high-level PHE-2AMI and the lower-level ABM-2AMI and PY-AMF⁵⁰. Therefore, the emission of TPAP-514 polymer can also exclude the contribution of TTET. In summary, the long-lived white light of TPAP-514 was proved to be only from the superposition of the triplet excited states of three components.”

In addition, the references cited in this part (Matter. 2023, 6, 217-225. Nat. Commun. 2019, 10, 1595) can be found in Ref. [49], [50].

Figure R6 (Figure 4). Luminescence properties of the white light-emitting TPAP-514 film. a Steady-state photoluminescence spectra of the TPAP-514 film excited at 365 nm. **b** Two-dimensional excitation–emission spectra of TPAP-514 film (excitation wavelength from 200 to 400 nm, emission wavelength from 400 to 750 nm). **c** The phosphorescence decay curves of 512 nm emission band of TPAP-514 film under 365 nm excitation in the temperature range of 80–280 K. **d** Temperature-dependent phosphorescence spectra of TPAP-514 films from 80 to 280 K ($\lambda_{\text{ex}} = 365$ nm). **e** Time-resolved phosphorescence emission spectra of PAABM, PAPHE, PAPY, TPAP-514 at room temperature under 365 nm excitation. **f** Absorption and emission spectra of three kinds of one-component polymer films at room temperature. **g** Excitation spectra of TPAP-514 film obtained at 512 and 610 nm under room temperature, respectively. **h** Emission spectra of prompt and delayed phosphorescence of TPAP-514 at different delay times ($t_d = 2$ –100 ms).

Figure R7 (Figure. 5d). White light-emission mechanism of different excited states. “Phos.” stands for phosphorescence.

2. The precise structures shown for multi-components copolymers should be confirmed by convincing experimental data. The one-pot reaction for multi-components copolymerization would NOT be 100% to obtain pre-ratios copolymers. The materials obtained in this work were actually mixtures, so ‘intrinsic’ RTP might mislead readers. The authors should provide data about reproducibility.

Response: We appreciate the valuable comments from the reviewer on our work. According to your proposal, we resynthesized all three-component polymers and reprecipitated and purified them three times in ethyl acetate, and then supplemented the ^1H NMR spectra of all these polymers. The proton peaks on all benzene rings were in the range of 6.5-9.2 ppm. In all polymers with a three-component feed ratio of 811 to 118, the corresponding hydrogen proton peak area on the pyrene ring decreases and the corresponding hydrogen proton peak on the PHE-2AMI ring

increases (Figure R8). Quantitatively, we take TPAP-811 and TPAP-118 as examples (Figures R9 and R10) (Supplementary Figures 10 and 15), the double peak at 7.35 ppm is attributed to the hydrogen proton peak of the clockwise 3-position carbon on the pyrene ring, the hydrogen proton peak at 9.1 ppm comes from the carbon adjacent to the N atom on PHE-2AMI. It is worth noting that the number of hydrogen atoms here is 2, the single peak at 6.95 ppm is from the hydrogen proton peak of a single carbon on the benzene ring of ABM-2AMI. Therefore, we integrate these characteristic peaks with the 12.2 ppm carboxyl hydrogen proton peak, and calculate that in TPAP-118, the integral area ratio of the four units is 260 : 1 : 1 : 6. After the number of hydrogen atoms is distributed, the actual ratio of the polymer is $n : m : x : y = 260 : 1 : 1 : 3$. Similarly, in TPAP-811, the exact structure can be determined to be $n : m : x : y = 250 : 3 : 0.5 : 0.5$. This is close to our pre-ratio. In addition, the ^1H NMR spectra of all the other three-component polymers and the specific structures confirmed can be found in the revised version of the Supplementary Figures 11-14. Therefore, according to your suggestion, on page 13, we modify the description as follows:

“In this study, the molar ratio of ABM-2AMI:PAA-NHS was fixed at 1, following with the PY-AMI and PHE-2AMI from 1:8 (TPAP-118) to 8:1 (TPAP-811), the precise structure of the three-component polymers was confirmed by ^1H NMR spectra (Supplementary Figures 10–15). And a blue-to-pink afterglow of approximately 3 s was noticed after the 365 nm UV excitation source was turned off.”

In addition, for the second problem, our material is a micro-crosslinked intrinsic polymer obtained by covalently coupling PAA-NHS precursors with different chromophores. On the one hand, the above ^1H NMR spectra results can prove the structure of these polymers. On the other hand, taking TPAP-316 as an example, we tested the phosphorescence spectra of the polymer after different reprecipitation times in ethyl acetate (Figure R11). The results showed that the phosphorescence properties of the polymer remained basically unchanged after multiple purifications, which could exclude the influence of impurities on the phosphorescence of these

materials. This proves that these polymers are different from the mixture of doping systems. In fact, the term 'intrinsic' is used to describe polymers to distinguish doped polymer systems. The above data indicated that the three-component polymer in this work is a slightly crosslinked intrinsic polymer. According to your suggestion, we will make the description more clear. We will refer to the intrinsic polymer room temperature phosphorescence referred to as 'IPRTP' for the first time, and replace all 'PRTP' in the full text with 'IPRTP'.

Figure R8. Hydrogen proton peak corresponding to benzene ring in three-component polymer.

Figure R9 (Supplementary Figure 10). ^1H NMR spectra of pure TPAP-118 in $\text{DMSO-}d_6$.

Figure R10 (Supplementary Figure 15). ¹H NMR spectra of pure TPAP-811 in DMSO-*d*₆.

Figure R11. Phosphorescence emission spectra of TPAP-316 after different times of reprecipitation.

3. In addition, so-called ‘applications of the obtained polymers in light-emitting diodes’ shown in Fig. 6 were still photo-excited RTP polymers films and were NOT real ‘OLED’ devices with electric-driven (such RTP polymers acted as active layer).

Response: We greatly thank the reviewer for raising the question. In fact, these multi-color polymer RTP material is indeed applied to the surface of the LED lamp in the form of a thin film as a ‘afterglow lampshade’, not the active layer in the ‘OLED’ device. Our application principle is still photoluminescence. In order to avoid the misunderstanding of the readers, we have redrawn the schematic diagram in Figure R12 (Fig.6) to clearly show the film properties of the polymer for LED applications. In addition, on page 21, the description of the application section has been modified as follows:

“After revealing the long-lived RTP mechanism of amorphous polymers, we explored the potential applications of the prepared materials in the field of light-emitting diodes. The single-component and three-component polymers were attached to the surface of UV LED ($\lambda_{ex} = 365$ nm) in the form of films by spin coating to fabricate a film lampshade (Fig. 6a). The LED assembled with the film lampshade and unassembled LED together formed an array that emitted bright luminescence (Fig. 6b). After the power supply was cut off, the LEDs applied to the film lampshade exhibited attractive multicolor afterglow, which were demonstrated as “C”, “Q”, “U”, and “T” patterns. Interestingly, by programmatically regulating the external circuit of the LED array, the afterglow exhibited different display paths. Additionally, the variable driving voltage endowed the three-component white afterglow film LED assemblies with different emission efficiencies (Fig. 6c). Because of its excitation and emission characteristics, PAACR was applied to dynamic anti-counterfeiting. PAACR solution was solidified in a mold to obtain the dry letter pattern “PRTP” (Fig. 6d). Under visible-light excitation, similar to the case of exposure to natural light, the pattern demonstrated the original color of the polymer; in contrast, under 365 nm UV excitation, the pattern exhibited bright blue fluorescence (Fig. 6e). After removing the two excitation sources, both patterns demonstrated yellow RTP with different intensities.”

Figure R12 (Figure 6). Application of polymeric RTP materials in light-emitting diodes (LEDs) and dynamic anti-counterfeiting. **a** Schematic of the synthesis of LED afterglow lampshade. **b** One component polymeric RTP material for LED display. DC on and off represent the switching on and off of power supply, respectively. The arrow indicates the direction of the afterglow path. ‘S’ stands for the starting point and ‘T’ stands for the end point. **c** Images of three-component white light emitter for afterglow lampshade under different driving voltages. **d** Processing schematic of PAACR for dynamic anti-counterfeiting. **e** Luminescence images of the one-component polymer model material PAACR under natural light, white light flash excitation, and 365 nm ultraviolet light excitation. VL denotes visible light.

4. In fact, polymeric room-temperature phosphorescence (RTP) materials had been reported by many scientists (Prof. Z. Li with many works e.g. Refs. [1, 19], Prof. Y-L. Zhao with many works e.g. Ref. [7], Prof. Marder with Ref. [10], Prof. Ogoshi with Ref. [22], Prof. Z. An with many works e.g. Ref. [30], Prof. X. Ma with many works e.g. Recent Advances of Pure Organic Room Temperature Phosphorescence Based on Functional Polymers. Accounts Materials Research, 2023, 10.1021/accountsmr.3c00090). Therefore, this work lacks novelty and could NOT be accepted for publication in Nature Comm.

Response: Many thanks for your comments. In recent years, long-lived persistent luminescence of polymers has been a research hotspot in the field of luminescent materials. However, it is also very rare to achieve multicolor, especially white light emission in non-doped polymer systems. Therefore, we believe that this work will be of great interest to readers in the field of chemistry and luminescent materials. Here, we would like to re-clarify the main novelties of this work:

(1) A series of long-lived polymer luminescent films were prepared by a simple and convenient one-pot reaction and drop-coating method without involving complex assembly modes.

(2) By covalently introducing three chromophores into the polymer precursor in different proportions, the obtained multi-component micro-crosslinked polymer achieves a wide range of afterglow color tunable from blue to red. In particular, copolymer TPAP-514 exhibits a rare white light emission in intrinsic polymers.

(3) Compared with other reported works on achieving white light emission, in our work, through comprehensive comparison, photophysical analysis and theoretical calculation, the possible singlet excited states and the interference of TTET on the emission of the three-component polymer are excluded, and it is confirmed that the white light emission comes from different triplet excited states. We believe that this work provides a new strategy for the development of white afterglow polymer system.

(4) Due to the multi-color long-life luminescence characteristics of these polymer system and the easy film-forming characteristics, it is used as an afterglow LED film lampshade to broaden its application scenarios.

As for the previous research in the field of polymer RTP proposed by the reviewers, it is undoubtedly very exciting. However, they would not show negative impact on the novelty of this work, because there are many differences between these previous works and our work. To clearly demonstrate the differences and the developed novelties of our work, we would like to do some comparisons with the mentioned literatures, however, this does not mean we aim to undermine the beautiful works of other scientists.

Figure R13. Multistage Stimulus-Responsive Room Temperature Phosphorescence Based on Host-Guest Doping Systems.

Angew. Chem. Int. Ed., 2021, 60(37), 20259-20263: This work reports a host-guest doping system with stimuli-responsive RTP characteristics based on triphenylphosphine oxide as the host and

benzo (dibenzo) phenothiazine dioxide derivatives as the guest. However, these materials are composed of small molecules and do not involve polymers, so they cannot reflect the good processability and other properties of polymers. In addition, the system focuses on the realization of multi-level stimulus response from grinding to chemical stimulation, and phosphorescence only shows a single green emission.

Figure R14. Ultraviolet irradiation-responsive dynamic ultralong organic phosphorescence in polymeric systems.

Nat. Commun, 2021, 12(1), 2297: In this work, a series of UV-responsive IRRTP systems were developed by doping eight phosphors into the PVA matrix. After continuous irradiation for 45 min, these materials exhibit irradiation-enhanced phosphorescence emission, but they are all physically doped systems, and it is difficult to avoid problems such as phase separation. At the same time, the afterglow color is only green and yellow.

Figure R15. Stimulus-responsive room temperature phosphorescence materials with full-color tunability from pure organic amorphous polymers.

Sci. Adv, 2022, 8(8), eabl8392: The highlight of this work is to use water as a solvent to achieve water/thermal stimuli-responsive full-color tunable RTP through the covalent binding of PVA with five kinds of arylboronic acids, and apply it to multi-layer information encryption and multi-color paper ink. However, the afterglow color in this system is related to the excitation wavelength, and white light emission cannot be achieved.

Figure R16. Ultralong room-temperature phosphorescence from amorphous polymer poly(styrene sulfonic acid) in air in the dry solid state.

Adv. Funct. Mater., 2018, 28(16), 1707369: In this work, the RTP lifetime is adjusted by adjusting the molecular weight of poly(styrene sulfonic acid) (PSS) and the ratio of sulfonic acid groups. The highlight is that the maximum lifetime of the obtained amorphous polymer reaches 1.22 s, which is three times that of the previous work. However, the system only shows pure green afterglow emission, and there is no report on color adjustment.

Figure R17. Circularly polarized organic room temperature phosphorescence from amorphous copolymers.

J. Am. Chem. Soc, 2021, 143(44), 18527-18535: In this work, axially chiral chromophores were incorporated into polyacrylic acid chains by free radical cross-linking polymerization, and circularly polarized organic phosphorescence with CPP efficiency up to 30.6 % was obtained in amorphous polymers. However, these materials only show yellow and green afterglow emission, and a wider color tunable range has not been reported.

In addition, in the reviewer 's summary of Prof. X. Ma's review of the research progress of *Pure Organic Room Temperature Phosphorescence Based On Functional Polymers (Accounts Materials Research, 2023, 10.1021/accountsmr.3c00090)*, the realization of white light emission in non-doped polymer systems is also extremely rare. We have re-cited the article in Ref. [25]. In another work (*CCS Chem. 2022, 4(1), 173-181*), Prof. X. Ma and his colleagues copolymerized two monophenyl compounds with acrylamide, and realized white light in the polymer system by

constructing triplet to singlet Förster resonance energy transfer. It is worth noting that the white light emission in this work includes the fluorescence emission generated by the phosphorescence energy transfer from the phosphorescence emission at the low band (425 nm). This is essentially different from our work. By introducing three chromophore groups into the polymer precursor, we rely on the superposition of the three independent triplet excitons to form a phosphorescence emission to form white light. In addition, the afterglow color adjustable region extends to blue to red. We have also confirmed that the emission comes from the triplet excited state of the three components.

The relevant literature (Acc. Mater. Res. 2023, 4, 827-838, CCS Chem. 2022, 4(1), 173-181) has been cited in Ref. [25], [38].

Figure R18 (CCS Chem. 2022, 4(1), 173-181). (a) Molecular structures of P1–P10. (b) Energy diagram (based on the optimized geometry of T_1 state) of the excited states S_1 and T_n .

Most importantly, we summarize some of recent reports on the construction of white light emission in organic systems. These reports include small molecules and polymer materials. In Figure R19, the triangle represents that these works mainly construct white light in mixed singlet

and triplet states. The diamond indicates that the emission comes from intramolecular energy transfer or charge transfer, and the pentagonal star means white light caused entirely by the triplet state. In summary, the white long-lived RTP of triplet excited states in undoped polymer systems is rare in previous studies. Our work provides a new strategy for the development of polymer white RTP system.

Figure R19. Some reports on organic white light emission in recent years. "S & T" (triangle) represents the emission from the mixture of singlet and triplet states; "ET & CT" (diamond) represents energy transfer or charge transfer; "T" (pentagonal star) represents the emission from the triplet state completely.

Reference

1. Xu, B. et al. White-light emission from a single heavy atom-free molecule with room temperature phosphorescence, mechanochromism and thermochromism. *Chem. Sci.* **8**, 1909-1914 (2017).
2. He, Z. et al. White light emission from a single organic molecule with dual phosphorescence at room temperature. *Nat. Commun.* **8**, 416 (2017).
3. Wang, J. et al. A facile strategy for realizing room temperature phosphorescence and single molecule white light emission. *Nat. Commun.* **9**, 2963 (2018).
4. Jinnai, K., Kabe, R. & Adachi, C. Wide-range tuning and enhancement of organic long-persistent luminescence using emitter dopants. *Adv. Mater.* **30**, 1800365 (2018).
5. Wang, X. et al. Pure organic room temperature phosphorescence from excited dimers in self-assembled nanoparticles under visible and near-Infrared irradiation in water. *J. Am. Chem. Soc.* **141**, 5045-5050.
6. Wen, Y. et al. Achieving highly efficient pure organic single-molecule white-light emitter: the coenhanced fluorescence and phosphorescence dual emission by tailoring alkoxy substituents. *Adv. Opt. Mater.* **8**, 1901995 (2020).
7. Kuila, S. et al. All-organic, temporally pure white afterglow in amorphous films using complementary blue and greenish-yellow ultralong room temperature phosphors. *Adv. Funct. Mater.* **30**, 2003693 (2020).
8. Roy, B. et al. Mapping the regioisomeric space and visible color range of purely organic dual emitters with ultralong phosphorescence components: from violet to red towards pure white light. *Angew. Chem. Int. Ed.* **61**, e20211805 (2022)
9. Barman, D., Annadhasan, M., Chandrasekar, R. & Iyer, P. K. Hot-exciton harvesting via through-space single-molecule based white-light emission and optical waveguides. *Chem. Sci.* **13**, 9004-9015 (2022).

10. Chong, K. et al. Structurally resemblant dopants enhance organic room-temperature phosphorescence. *Adv. Mater.* **34**, 2201569 (2022).
11. Gui, H., Huang, Z., Yuan, Z. & Ma, X. Ambient white-light afterglow emission based on triplet-to-singlet Förster resonance energy transfer. *CCS Chem.* **4**, 173-181 (2022).
12. Jin, J. et al. Modulating tri-mode emission for single-component white organic afterglow. *Angew. Chem. Int. Ed.* **60**, 24984-24990 (2021).
13. Yang, Z. et al. Boosting the quantum efficiency of ultralong organic phosphorescence up to 52% via intramolecular halogen bonding. *Angew. Chem. Int. Ed.* **59**, 17451–17455 (2020).
14. Ma X, et al. A color-tunable single molecule white light emitter with high luminescence efficiency and ultra-long room temperature phosphorescence. *J. Mater. Chem. C.* **9**, 727-735 (2021).
15. Liang, X., Zheng, Y. & Zuo, J. L. Two-photon ionization induced stable white organic long persistent luminescence. *Angew. Chem. Int. Ed.* **60**, 16984-16988 (2021).

Special thanks to your good comments and suggestion again! We sincerely hope that the revised manuscript is now suitable for publication in the journal.

Reviewer #2 (Remarks to the Author):

In this work, Yang and co-authors present a strategy of covalently coupling different conjugated chromophores with poly(acrylic acid (AA)-AA-N-succinimide ester) by a simple and rapid one-pot reaction to obtain slightly crosslinked polymers with long-lived RTP of various colors. Interestingly, the afterglow colors of polymers can be modulated from blue to red by introducing three chromophores into them, the acquired polymer TPAP-514 exhibited pure white afterglow at room temperature with the chromaticity coordinates (0.33, 0.33). Moreover, the potential applications of the obtained polymers in light-emitting diodes and dynamic anti-counterfeiting are explored. I think the proposed strategy provides a new idea for constructing intrinsic polymers with diverse RTPs. Based on the rational material preparation and the outstanding optical performance, it can be recommended for publication in Nature Communications after minor revision.

Response: Thank you for your recognition of this work and the giving follow constructive comments. According to those comments, we have carefully revised the relative parts in our manuscript. Below are our point-to-point responses to the reviewer's comments.

1. In this work, only ^1H NMR and FTIR were used to characterize the polymers. ^{13}C NMR spectra of PAPHE, PAABM, PAACR, PAPY is recommended to be used for further demonstration of their chemical structures.

Response: We thank the reviewer for this valuable advice. The corresponding ^{13}C NMR spectra data of each polymer were supplemented in **Figures R20-23 (Supplementary Figures 6-9)**. It is worth mentioning that due to the poor solubility of the micro-crosslinked polymer PAPHE in deuterated DMSO, the peaks in the liquid ^{13}C NMR results are not obvious. Therefore, we use its solid nuclear magnetic ^{13}C spectrum instead.

Figure R20 (Supplementary Figure 6). ^{13}C NMR spectra of pure PAACR in $\text{DMSO-}d_6$.

Figure R21 (Supplementary Figure 7). ^{13}C NMR spectra of pure PAABM in $\text{DMSO-}d_6$.

Figure R22 (Supplementary Figure 8). ^{13}C NMR spectra of pure PAPY in $\text{DMSO-}d_6$.

Figure R23 (Supplementary Figure 9). Solid-state ^{13}C NMR spectra of PAPHE.

2. How did you confirm the long-lived luminescence of polymers from RTP rather TADF?

Response: We thank the reviewer for raising this question. As we know, temperature-dependent spectra is a feasible method to distinguish TADF from RTP. In Figure 4d, Figures R24 and R25 (Supplementary Figures S34 and 35), we show that as the temperature increase from 80 K to 280 K, the intensity of phosphorescence emission decreases significantly, and at the same time, the phosphorescence lifetime reduces due to thermal vibration makes triplet excitons more depleted in a non-radiative way. Therefore, it can be proved that the emission comes from RTP rather TADF in this work. Unfortunately, we realized that the temperature-dependent spectrum of PAACR was not given in original Supporting Information, so we added it to Figure R24 (Supplementary Figure S34).

Figure R24 (Supplementary Figure 34). a-d Phosphorescence spectra of PAPHE (a), PAABM (b), PAACR (c) and PAPY (d) films excited at 365 nm in the temperature range of 80-280 K.

Figure R25 (Supplementary Figure 35). a-d Phosphorescence decay curves of PAPHE (a), PAABM (b), PAACR (c) and PAPY (d) films excited at 365 nm in the temperature range of 80-280 K.

3. For polymer PAPHE, how did you separate fluorescence and phosphorescence spectra when calculated the quantum yield?

Response: Thank you for your good question. For polymer PAPHE, the emission peaks of fluorescence and phosphorescence spectra are very near, to separate them, we utilized the delay time when calculated phosphorescence quantum yield. To further confirm the correct of

phosphorescence quantum yield, we have recalculated the phosphorescence quantum yield of PAPHE at different delay times (0.5-5 ms) (Figure R26), and the results can be kept in a stable range, indicating the testing method of phosphorescence quantum yield is relatively reasonable.

Figure R26. Phosphorescence quantum yield of PAPHE at different delay time under environmental conditions.

4. The Figures 2 and 3 should be clearly displayed, some font size for Figure 2 should be increased.

Response: Thank you for your effective advice. According to your suggestion, we have clarified Figs. 2f and g, and increase the font size in Fig. 2a (Figure R27). In addition, Figure R28 (Fig. 3) has been remade, and you can view them in revised manuscript.

Figure R27 (Figure 2). Photophysical properties of single-component polymer films with room-temperature phosphorescences. a Images of long-lived RTPs of four single-component polymer films excited by a 365 nm UV flashlight. **b-e** Photoluminescence spectra of PAACR (**b**), PAPHE (**c**), PAABM (**d**), and PAPY (**e**) acquired under 365 nm excitation and ambient conditions. **f** Phosphorescence spectra of PAACR films with different chromophore concentrations under 365 nm excitation at room temperature. **g** Phosphorescence spectra and quantum yields of PAACR films with different acrylic acid contents under 365 nm excitation at room temperature. **h** Phosphorescence lifetimes and quantum yields of the four single-component polymer films under 365 nm excitation.

Figure R28 (Figure 3). Feasible strategy of three-component regulation of the RTP color of an amorphous polymer. **a** Images of the phosphorescences of the films with different proportions of three-component polymer under 365 nm excitation at room temperature. xyz represents the PY-AMI: ABM-2AMI: PHE-2AMI ratio. **b** Schematic diagram and local molecular formula of the three-component afterglow-tunable polymer TPAP. **c** Phosphorescence spectra of three-component polymers with different PYAMI: ABM-2AMI: PHE-2AMI ratios under 365 nm excitation and environmental conditions. **d** Phosphorescence lifetimes of the three-component polymers with different PY-AMI: ABM-2AMI: PHE-2AMI ratios at 500 nm and 610 nm under environmental conditions ($\lambda_{ex} = 365$ nm). **e** CIE coordinate diagram corresponding to the phosphorescence spectra of the three-component polymers.

5. The GPC data of PAABM, PAACR, PAPY should be provided.

Response: Thanks for your suggestion. According to your proposal, we have supplemented the GPC data of the corresponding polymer in **Figure R29 (Supplementary Figure 17)** and **Table R1 (Supplementary Table 1)**.

Figure R29 (Supplementary Figure 17). a-e GPC curves of polymers TPAP-514 (a), PAPHE (b), PAABM (c), PAACR (d) and PAPY (e) using DMSO as the mobile phase.

Table R1 (Supplementary Table 1). GPC data of polymers using DMSO as the mobile phase.

System	Mn	Mw	Mp	Mz	Polydispersity	Mz/Mw	Mz+1/Mw
TPAP-514	40700	95160	81055	174506	2.34	1.83	2.79
PAPHE	44248	103408	88287	189813	2.34	1.84	2.80

PAABM	18747	41250	35036	76097	2.20	1.84	2.98
PAACR	20891	44696	39376	79067	2.15	1.77	2.75
PAPY	16331	38447	33735	71925	2.35	1.87	3.05

6. In Fig. S13, S14, the TGA and DSC curves of all polymers PAPHE, PAABM, PAACR, PAPY should be added.

Response: Many thanks for your valuable advice, we have added the TG and DSC curves of all polymers to Figures R30 and 31 (Supplementary Figures 19 and 20).

Figure R30 (Supplementary Figure 19). Thermogravimetric analysis curves of PAA-NHS and all single-component polymers with heating rate of 10 °C/min under the N₂ atmosphere.

Figure R31 (Supplementary Figure 20). a-f DSC curves of PAA-NHS (a), PAPHE (b), PAABM (c), PAACR (d), PAPHY (e) and TPAP-514 (f).

Special thanks to your good comments and suggestion again!

Reviewer #3 (Remarks to the Author):

In this work, the author reported a one-pot reaction to covalently introduce conjugated chromophores into poly(acrylic acid (AA)-AA-N-succinimide ester) (PAA-NHS) to construct a micro-crosslinked long-lived intrinsic polymer RTP system. The intrinsic polymer PAPHE obtained by their strategy has a phosphorescence quantum yield of 14.7 %. Importantly, the polymer film achieves a wide range of color-tunable afterglow from blue to red including white light by rationally adjusting the feed ratio of the phosphorescent unit through multiple components. The multi-color long-life luminescence properties based on these polymers have also been reasonably applied to LED and anti-counterfeiting fields. This work has a great contribution to the development of polymer RTP field, especially provides a novel strategy for the development of white light RTP system. I think this manuscript will arouse the widely interest of readers of Nature Communications, so it can be recommended to publish after minor revision.

Response: We thank the precious time of the reviewer devoted to the reviewing process. We appreciate the reviewer for the evaluation of our manuscript. We have done our best to improve the manuscript according to the reviewer's suggestions.

1. To further confirm the chemical structure of desired polymer, ^{13}C NMR data should be added in revised manuscript.

Response: Thanks so much for your good advice. The corresponding ^{13}C NMR spectral data of each polymer were supplemented in **Figures R32-35 (Supplementary Figures 6-9)**. It is worth mentioning that due to the poor solubility of the micro-crosslinked polymer PAPHE in deuterated DMSO, the peaks in the liquid ^{13}C NMR results are not obvious. Therefore, we use its solid nuclear magnetic ^{13}C spectrum instead.

Figure R32 (Supplementary Figure 6). ^{13}C NMR spectra of pure PAACR in $\text{DMSO-}d_6$.

Figure R33 (Supplementary Figure 7). ^{13}C NMR spectra of pure PAABM in $\text{DMSO-}d_6$.

Figure R34 (Supplementary Figure 8). ^{13}C NMR spectra of pure PAPY in $\text{DMSO-}d_6$.

Figure R35 (Supplementary Figure 9). Solid-state ^{13}C NMR spectra of PAPHE.

2. In Fig. 2h, the authors mentioned that the micro-crosslinked PAABM and PAPHE polymer films stimulated the construction of a rigid environment, but why PAACR exhibited longer phosphorescence (395.9 ms)?

Response: Thanks a lot for your professional question, although micro-crosslinking environment can effectively enhance the rigid environment and suppress non-radiative transitions. However, the RTP emission depends more on the degree of molecular conjugation and energy level matching. Obviously, the ACR-AMI molecule has a stronger degree of conjugation and is more likely to form π - π stacking, so we think it is relatively reasonable for its polymer PAACR to have a longer lifetime.

3. In Figure 3d, due to the existence of multiple emission centers, the multi-component polymer should exhibit different phosphorescence lifetimes in different emission bands. Therefore, the

author should confirm whether the collected phosphorescence lifetime comes from the same emission band, because there is no annotation here.

Response: Thanks for your good suggestion, we are sorry that the description of the phosphorescence lifetime of the three-component polymer is not accurate enough. We have remeasured the lifetimes of TPAP-514 at 500 nm and 610 nm and supplemented them in **Figure R36 (Figure 3d)**.

Figure R36 (Figure 3d). Phosphorescence lifetimes of the three-component polymers with different PY-AMI: ABM-2AMI: PHE-2AMI ratios at 500 nm and 610 nm under environmental conditions ($\lambda_{ex} = 365$ nm).

4. In the application as a afterglow LED lampshade, the picture is not particularly clear, and in Figure 6b, what is the meaning of “S”, “T”?

Response: Thank you for your question, we have clarified Figure 6 (Figure R37) so that the application can be displayed more clearly and intuitively. “S” and “T” are the starting point and end point of the afterglow path. “S” represents the starting point and “T” represents the terminal point. We have supplemented this part of the explanation in the figure note.

Figure R37 (Figure 6). Application of polymeric RTP materials in light-emitting diodes (LEDs) and dynamic anti-counterfeiting. **a** Schematic of the synthesis of LED afterglow lampshade. **b** One component polymeric RTP material for LED display. DC on and off represent the switching on and off of power supply, respectively. The arrow indicates the direction of the afterglow path. ‘S’ stands for the starting point and ‘T’ stands for the terminal point. **c** Images of

three-component white light emitter for afterglow lampshade under different driving voltages. **d** Processing schematic of PAACR for dynamic anti-counterfeiting. **e** Luminescence images of the one-component polymer model material PAACR under natural light, white light flash excitation, and 365 nm ultraviolet light excitation. 'VL' denotes visible light.

5. The color of annotation of figures S16, S17 and S20 should be black.

Response: Thanks to the reviewer 's kind advice, we have modified the annotations of the original S16, S17 and S20 (Supplementary Figures 22, 23 and 26).

Special thanks to your good comments and suggestion again!

REVIEWERS' COMMENTS

Reviewer #2 (Remarks to the Author):

In the previous version, reviewer #1 was mainly concerned about the nature of emission, precise polymer structures, and potential applications. In the revised manuscript, the authors have conducted a series of experiments to give sufficient verification and response. The experimental phenomena and conclusions are mutually supportive. Additionally, the concern about the novelty of this topic is overdone. As the authors point out, the previous reports have little impact on the novelty of this work. The authors have addressed the reviewer's issues, including my concern. Therefore, I believe this work deserves to be published in Nature Communications as it is.

[Note from the Editor: Reviewer #2 was asked to look also over the response given to Reviewer #1]

Reviewer #3 (Remarks to the Author):

This version can be accepted.

Reviewer #2 (Remarks to the Author):

In the previous version, reviewer #1 was mainly concerned about the nature of emission, precise polymer structures, and potential applications. In the revised manuscript, the authors have conducted a series of experiments to give sufficient verification and response. The experimental phenomena and conclusions are mutually supportive. Additionally, the concern about the novelty of this topic is overdone. As the authors point out, the previous reports have little impact on the novelty of this work. The authors have addressed the reviewer's issues, including my concern. Therefore, I believe this work deserves to be published in Nature Communications as it is.

Response: We appreciate you for your kind recommendation, and thank you for the time and energy spent in the review process.

Reviewer #3 (Remarks to the Author):

This version can be accepted.

Response: Thank you for your kind recommendation and the effort for improving the quality of our manuscript.